# Beyond Trend and Periodicity: Guide Time Series Forecasting with Textual Cues

## Abstract

This work introduces a novel **Text-Guided Time Series Forecasting** (TGTSF) task. By integrating textual cues, such as channel descriptions and dynamic news, TGTSF addresses the critical limitations of traditional methods that rely purely on historical data. To support this task, we propose *TGForecaster*, a robust baseline model that fuses textual cues and time series data using cross-attention mechanisms. We then present four meticulously curated benchmark datasets to validate the proposed task, ranging from simple periodic data to complex, event-driven fluctuations. Our comprehensive evaluations demonstrate that TGForecaster consistently achieves state-of-the-art performance, highlighting the transformative potential of incorporating textual information into time series forecasting. This work not only pioneers a novel forecasting task but also establishes a new benchmark for future research, driving advancements in multimodal data integration for time series models.

## 1 Introduction

Time series forecasting (TSF) is crucial in many fields, thereby attracting significant attention from both academia and industry. Despite extensive research dedicated to this task, recent studies have shown that simple linear models (Zeng et al., 2023; Xu et al., 2023; Toner & Darlow, 2024) that barely extract trend and periodicity information from time series data frequently achieve performance close to that of state-of-the-art complex models (Nie et al., 2023; Liu et al., 2023; Jin et al., 2023), indicating that current approaches have reached a saturation point, possibly overfitting historical data.

Beyond simple trends and periodicity, complex patterns in time series data are often influenced by external factors like holidays and consumer sentiment—information not captured by historical data alone. To improve forecasting accuracy, it is essential to incorporate such external information as inputs. Otherwise, models may fail to identify key causal relationships, leading to oversimplification or overfitting. Previous works (Wu et al., 2021; Zhou et al., 2022a; Rasul et al., 2024) have explored translating simple information into auxiliary channels, such as using one-hot encoding, but this approach often loses valuable semantic information, reducing the model's ability to generalize and extract meaningful insights. Additionally, the lack of system dynamics and inter-channel relationships can lead to unreliable forecasts. Since much of this information is text-based and difficult to express as time series data, it is crucial to develop models that can understand and integrate textual inputs. Although some have attempted to leverage large language models (LLMs) for time series forecasting (Cao et al., 2023; Jin et al., 2023), recent studies (Tan et al., 2024) indicate that LLMs still face challenges in aligning token-based methods with the temporal domain. This underscores the need for a multi-modal approach that effectively combines text and time series data.

We propose a novel multi-modality task: **Text-Guided Time Series Forecasting (TGTSF)**. TGTSF leverages two extra text-based components: channel descriptions and dynamic news messages. Channel descriptions provide static knowledge about the underlying systems, enabling the model to differentiate between channels and better understand inter-channel correlations. News messages offer dynamic and external insights, helping the model adjust to shifts in data distribution caused by external events. By modeling the joint distribution of future values conditioned on these textual cues, TGTSF has the potential to enhance forecasting accuracy and reliability. Furthermore, TGTSF enables scenario-based forecasting, which is particularly valuable for business decision-making. For

instance, a company can forecast sales based on the assumption of a successful marketing campaign, incorporating relevant news and channel-specific descriptions to refine predictions.

We introduce **TGForecaster**, a Transformer-based multimodal model designed to effectively leverage textual information for time series forecasting. TGForecaster incorporates two key innovations: **Time-Synchronized Text Embedding** and **Text-Guided Channel Independent (TGCI)**. Instead of word token, we use sentence embedding vector to carry the semantic information. By ordering these vectors on time domain, news messages are aligned with their corresponding time steps, allowing the model to incorporate relevant external information at the right moments. The TGCI mechanism dynamically attends to the most relevant news for each specific time series channel, enabling the model to capture the unique impact of different news items on various channels.

To validate this task, we propose four multimodal datasets as a benchmark, each designed to test different aspects of the model's capabilities. Specifically, according to the real-world applications, we consider three categories news messages. (1) Common knowledge, such as dates and public events, to provide foundational context and align the model with predictable patterns; (2) system-level limited predictions, which incorporate sparse and broad domain knowledge, like weather or market reports, to guide the model in generating accurate and fine-grained time series predictions; and (3) hypothesized or controlled events, allowing for "what-if" scenario analysis, where planned actions, such as marketing campaigns, can potentially influence future outcomes. These components are derived from real-world scenarios and are known before forecasting begins, ensuring the benchmark is both leakage-free and valid.

Experiment results show that TGForecaster consistently demonstrates state-of-the-art performance across our datasets, affirming the soundness of the TGTSF task definition. Ablation studies reveal that without textual assistance, TGForecaster's performance reverts to that of PatchTST (Nie et al., 2023), its time series encoder backbone, underscoring that the performance enhancement is driven by the additional information provided by textual data, not merely by a more sophisticated architecture.

Our contributions are summarized as follows:

- We identify the roadblock of TSF as information insufficiency and propose TGTSF as a new forecasting approach that integrates textual data to enrich the models with external causal information and system knowledge.
- We establish the first TGTSF benchmark containing four uniquely designed datasets.
- We design a simple baseline model for TGTSF, TGForecaster. TGForecaster achieves state-of-the-art performance by utilizing textual information and effectively validates our proposed TGTSF task.

## 2 INSIGHTS AND MOTIVATION

### 2.1 EXISTING TSF SOLUTIONS SUFFER FROM INFORMATION INSUFFICIENCY

Time series forecasting typically involves predicting future segments based on historical data, specifically forecasting subsequent time series segments from prior ones. However, the inherent sparsity of information within time series data makes achieving accurate forecasts challenging, or even impossible at times. Standard decomposition of time series data identifies three main components: trend, periodicity, and noise (Box et al., 2015). The noise component is intrinsically unpredictable. Trends, which are slowly changing patterns, are often impacted by external events that influence the underlying system, making it less predictable without external information, see Appendix C for detailed discussion. As a result, the periodicity component is usually the main source of reliable prediction.

Recent advancements in linear models have validated these challenges. For instance, the DLinear (Zeng et al., 2023) model utilizes a simple linear layer to effectively capture basic periodic patterns, outperforming most complex Transformer-based models. Similarly, FITS (Xu et al., 2023), designed to simply extract periodicity, achieves comparable or superior results to state-of-the-art (SOTA) methods using fewer than $10k$ parameters. Further research demonstrates that employing a closed-form forecasting matrix (Toner & Darlow, 2024), calculated from training data, can produce SOTA outcomes. These findings indicate that the minimal information in time series data allows models to recognize patterns with few or no learnable parameters, suggesting that training larger models with insufficient information can lead to severe overfitting.

The issue of information insufficiency extends beyond the data itself to the absence of external information. External factors like holidays, consumer sentiment, or climate changes can significantly alter patterns in time series data, affecting variables such as electricity usage, sales trends, or regional weather conditions. However, these factors are not contained within the time series data, hindering models from incorporating causal relationships. This deficiency can compromise model training, causing models to either learn overly simplified *average shortcut* or overfit on the training data, as shown in Fig. 1. For example, if a dataset frequently presents days without rain, the model may consistently predict no rain for future days. However, rainfall predictions are largely influenced by external factors like climate change, not included in the time series, leading models to opt for *shortcuts* that minimize overall loss by predicting the most common outcome.

Moreover, the lack of knowledge about the underlying systems exacerbates the problem of information insufficiency, especially in datasets with multiple time series channels. Without a deeper understanding of these systems or the characteristics of each channel, it is challenging to leverage such information for improved modeling performance. The indistinctiveness among channels complicates modeling efforts, making it difficult to accurately capture and model correlations.

These challenges underscore the need to integrate external information to address the fundamental limitations of time series forecasting methods.

## 2.2 RIN & WEIGHT SHARING ARE COMPROMISES UNDER INFORMATION INSUFFICIENCY

Researchers have developed various methods to tackle information insufficiency in time series forecasting. One such method is Reversible Instance Normalization (RIN) (Kim et al., 2021), which normalizes each instance to align mean and variance across the dataset, as shown in Fig. 1. RIN effectively reduces distribution shifts caused by trends shifting or external events, forcing the model to learn a unified distribution. While it standardizes data inputs, RIN also strips away essential information such as trend intensity and amplitude variations, which diminishes the model's capacity to detect relative biases or amplitude shifts beyond the training scope. However, RIN merely hides the issue of information insufficiency by neutralizing the effects of external events.

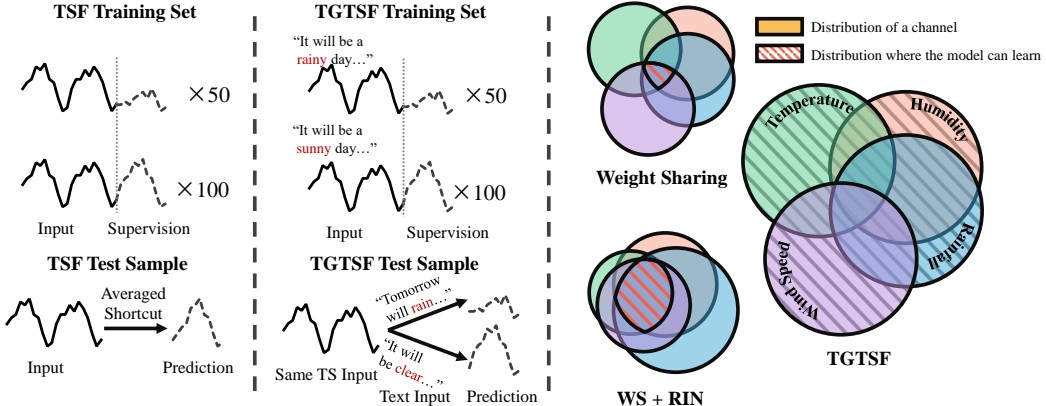

Figure 1: **Left Panel**: The traditional TSF model, when faced with distribution shifts due to external events, tends to learn an averaged shortcut for a smaller global loss, as shown in the leftmost figure. Conversely, the TGTSF task utilizes text inputs to eliminate the uncertainty and effectively generalize to these shifts, as depicted in the second figure. **Right Panel**: The rightmost figure illustrates the effect of text guidance on adapting to different channel distributions. Weight Sharing (WS) method helps the model capture only common knowledge among channels, while RIN addresses distribution shifts by normalizing instances, thus learning a larger portion of the distribution. TGTSF, however, leverages text to directly learn the entire distribution, enabling the model to generalize to a specific channel distribution with corresponding text as the condition.

Another strategy is weight sharing (Nie et al., 2023; Zeng et al., 2023; Xu et al., 2023). This technique treats all input channels as equivalent, leveraging common patterns across all channels to alleviate the problem of limited training samples. In other words, this approach enhances performance by learning from the shared periodicities across all the channels, typical within a single dataset. However, since the model is trained to minimize loss across all channel distributions with a single set of weights, it

prioritizes common patterns, failing to capture unique distributions for each channel, as illustrated in Fig. 1, which may lead to averaged results. These average results further reveal that the model lacks sufficient information to distinguish between different channels effectively.

Despite these advancements, both methods provide only temporary solutions to the pervasive issue of information insufficiency. They do not resolve the core problem and still suffer from the limitations imposed by insufficient information.

### 2.3 MOTIVATION FOR THE PROPOSED TGTSF FRAMEWORK

External factors significantly impact time series behavior such as trending and periodicity. A line of work apply and investigate how to embed the external information as a new channel, such as Autoformer (Wu et al., 2021), FedFormer (Zhou et al., 2022a) add auxilary channels to indicate the day-of-week or even the exact date. Temporal Fusion Transformer (TFT) (Lim et al., 2021) and TimeXer (Wang et al., 2024), however, focus on how to select the given variables and how to embed and fuse them with other channels. However, these methods yield limited performance improvements as they primarily extract more information from the already constrained time series dataset.

Incorporating additional information can reduce uncertainty and improve forecasting performance. Since much of this information is created for human consumption, it often comes in the form of text, making text a crucial modality for time series forecasting. However, integrating textual data as an external modality presents several challenges.

First, text is semantically rich and diverse, and reducing it to simple numerical forms, such as one-hot encoding, can result in significant loss of information, preventing the model from generalizing to unseen events and learning the causal relationships between events and time series patterns. Second, we aim to use the text to incorporate dynamic information and expert insights relevant to specific time stamps. Achieving precise temporal alignment between time series data and textual information remains challenging, particularly for word token-based methods like large language models (LLMs). Additionally, multiple news messages may correspond to a single time step, and different channels may react differently to the same information, adding complexity to the alignment process.

To address these challenges, it is crucial to explore a multi-modal framework for time series forecasting that integrates textual data alongside historical time series data. Such an approach would allow models not only to respond to past trends but also to proactively adapt to future events and trends. By incorporating textual information, this framework could help capture causal relationships and inter-channel dynamics that are often missing in traditional time series models. It would also enhance forecasting accuracy by providing insights into the underlying mechanisms that influence the time series, offering a more holistic view of the system. Thus, result in more accurate forecasting.

## 3 TGTSF TASK FORMULATION

We introduce a novel task within the domain of time series analysis, termed **Text-Guided Time Series Forecasting (TGTSF)**, designed to address the prevalent issue of information insufficiency in time series forecasting.

As illustrated in Fig. 1, text guidance in TGTSF operates on two aspects. Firstly, *channel descriptions* serve as identifiers for each channel, aiding the model in distinguishing between them while learning shared features. These descriptions can also incorporate *static* knowledge about the underlying system, enhancing the model's ability to recognize inter-channel correlations. Secondly, *news messages* provide *dynamic* and *external* insights into known (in training) or hypothesized (in inference) future events, which assist the model in adapting to event-driven distribution shifts. Specifically, to ensure data quality and prevent information leakage, we focus on the following three types of text information as news messages.

**Common knowledge** such as dates, public holidays, and notable events like Black Friday sales, aligns with the time series and is known in advance, so it can be safely incorporated into forecasting without causing information leakage.

**System-level limited predictions** from domain experts, like weather report or market analyses, offer valuable high-level domain insights not present in historical data. Although these predictions, based on expert knowledge, are *infrequent* and often expressed in *broad, imprecise* terms, they are available prior to forecasting and help guide the overall direction of predictions without data leakage.

**Hypothesized or controlled events** arise from planned actions within a system, such as forecasting sales after a marketing campaign or predicting user engagement after a software update. These insights are pre-existing and specific to the system, allowing for "what-if" scenario forecasting while avoiding information leakage.

Consider a weather forecasting model predicting rainfall patterns. First, **common knowledge** like the month, date, and time offers essential seasonal context—rainfall varies between seasons and times of day. This allows the model to align with broader seasonal trends without needing an extensive historical window. Second, **system-level limited predictions** from weather stations provide valuable insights, though these forecasts may be infrequent and vague, such as predicting the likelihood of rain without precise amounts. By correlating these atmospheric predictions with observed rainfall, the model can generate more precise time series forecasts at a higher resolution. Finally, **controlled events**, like artificial rainmaking, offer a unique input. Knowing in advance when and where such events will occur allows the model to adjust its predictions accordingly. Combining these three types of information enables the model to produce more accurate and context-aware forecasts.

In many multi-channel time series forecasting (TSF) tasks, news messages are not directly tied to specific time series channels, and each channel may react differently to the same news. Consequently, the model must infer the impact of each news item on individual channels based on their descriptions. For example, a forecast task involve predicting time series of rainfall, temperature, and air pressure. A news item about an approaching storm could directly influence the rainfall channel, while affecting temperature and air pressure to varying degrees. The model needs to learn these relationships to provide accurate predictions across all channels.

Thus, the TGTSF task is defined by three principal inputs: the *time series* data, *news messages*, and *channel descriptions*. This setup not only capitalizes on dynamic events but also integrates domain knowledge via channel descriptions. Such integration is crucial for the model to comprehend and learn the spatial correlations among channels, thereby enhancing the forecasting accuracy with nuanced contextual understanding.

Formally, the TGTSF task seeks to model the potentially complex joint distribution of future values in a multi-channel sequence. This modeling involves the integration of historical time series data, textual information from news messages, and channel descriptions over a specified look-back window:

$$P(X_{n:n+h}^j | (X_{n-L:n-1}^j, News_{n:n+h}, Des^j)), j \in [1, c] \tag{1}$$

where $L$ denotes the look-back window length, $h$ stands for the prediction length, the $j$ means the channel number, $c$ is the total number of channels.

## 4 TGFORECASTER: A BASELINE MODEL FOR THE TGTSF TASK

To validate the TGTSF task, we developed TGForecaster, a streamlined transformer model designed for multimodal fusion with cross-attention. Illustrated in Fig. 2, this model harnesses textual information to enhance the accuracy and relevance of time series forecasting. It employs a reimplemented PatchTST encoder for processing time series data and leverages off-the-shelf, pretrained text models for pre-embedding textual inputs into vector sequences across the time dimension, allowing for effective modality fusion in the embedding space.

The first key innovation of TGForecaster is its **Time-Synchronized Text Embedding** mechanism. Since events occur at specific time steps, it is crucial to align the corresponding news messages to those times. Using text embeddings, we can represent these news messages as uniform embedding vectors while retaining their semantic meaning. For each time step, we create a list of news message embeddings and stack these lists along the time dimension to form a news embedding tensor. To ensure valid tensor dimensions, each list is zero-padded to match the maximum number of news items across time steps.

Another novel component of TGForecaster is the integration of a cross-attention layer for **Text Guided Channel Independent (TGCI)** learning. This layer is essential for fusing news content with channel descriptions. Inspired by the attention mechanism in recommendation systems, we treat the news as the key and value, and the channel descriptions as queries. This allows the model to compute the relevance of each news item to every channel, generating a composite embedding for each channel based on the news.

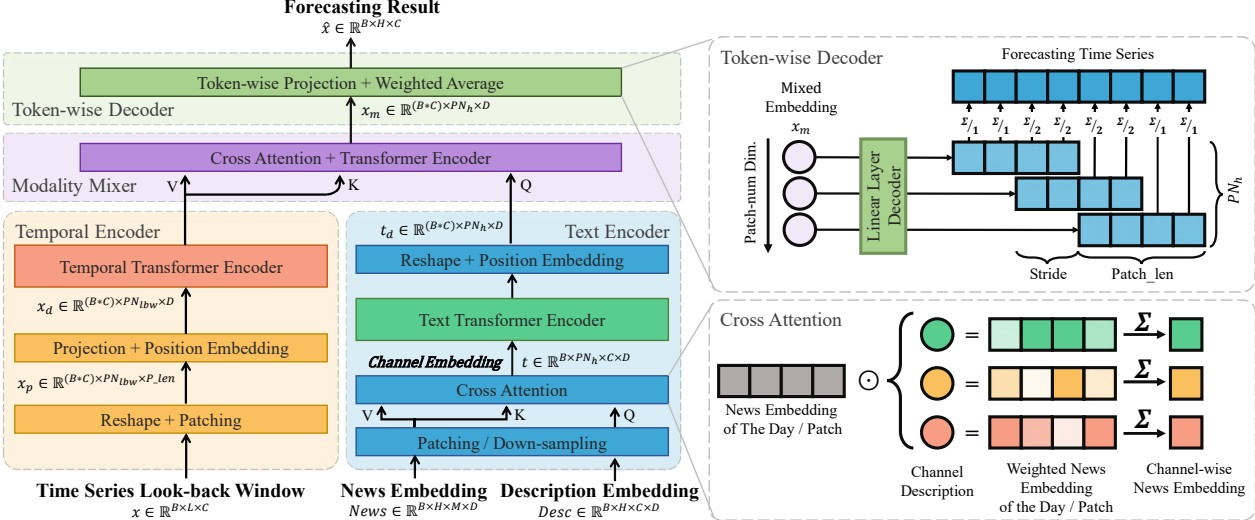

Figure 2: **Architecture of the TGForecaster Model.** TGForecaster integrates three primary inputs: time series data from a look-back window, news embeddings, and channel description embeddings. The texts are aligned to the forecasting horizon length. Text-side embeddings act as queries within the modality mixer layer to synchronize outputs with the forecast length. On the right panel, TGForecaster's efficiency is highlighted by the text encoder's cross-attention layer, which effectively associates news with specific channels. The token-wise decoder prevents overfitting in the final linear layer through inverse patching. Key parameters—number of patches ($PN$), maximum news items per batch ($M$), look-back window ($L$), and forecast horizon ($H$)—are streamlined for clarity.

Finally, we conduct modality fusion through another cross-attention layer, where text embeddings serve as queries, and temporal embeddings act as the key and value. This ensures that the output dimensions correspond to the forecasting horizon. The attention maps for both cross-attention layers, provided in Appendix H, clearly demonstrate that the model performs as designed.

As decoder, we use the token-wise decoding to avoid the overfitting on the last layer as observed in previous work (Lee et al., 2023). Each token is project back to its original patch length and weighted sumed to get the final forecasting result. Notably, TGForecaster does not incorporate the RIN. Our experiments, detailed in Appendix F, demonstrate that the inclusion of RIN degrades performance when sufficient information is already available.

This design strategy not only enables the integration of textual data into the time series forecasting model but also facilitates a direct comparison of the impact of text on forecasting accuracy.

## 5 TGTSF BENCHMARK DATASETS

We have designed four TGTSF benchmark datasets across three categories: synthetic, captioned existing, and real-world datasets. The datasets are designed as off-the-shelf temporally aligned time series-text multi-modal datasets. Each of them contains time series, channel description text and dynamic news.

### 5.1 SYNTHETIC TOY DATASET

The Synthetic Toy Dataset evaluates models' ability to utilize textual information for time series forecasting. It features diverse patterns, including segments of sinusoidal waves with variations in frequency, amplitude, and trend. Textual descriptions resembling news reports precede each change point, outlining upcoming alterations to prompt the TGTSF model to adjust its forecasts accordingly. This setup tests the model's effectiveness in mitigating distribution shifts and harnessing textual cues for improved forecasting accuracy.

### 5.2 ELECTRICITY-CAPTIONED DATASET

The Electricity-Captioned Dataset builds on a commonly-used electricity utility dataset (Zhou et al., 2021) that tracks appliance usage in an office building, with daily patterns largely influenced by whether it is a workday. This dataset has been enhanced with straightforward, **common knowledge**

textual information such as the type of day (e.g., day of the week, public holiday, workday), using the channel name as the descriptor. This enhancement serves a dual purpose: to illustrate whether minimal textual data can improve prediction accuracy and to facilitate a direct performance comparison with traditional time series models.

## 5.3 WEATHER-CAPTIONED DATASET

The Weather-Captioned Dataset is designed to overcome the limitations of commonly used datasets in time series forecasting (TSF), especially the lack of predictability and periodicity in variables like rainfall, wind speed, and direction. By incorporating external **limited system-level predictions**, this dataset demonstrates how to generate fine-grained time series prediction according to coarse-grained textual human prediction.

Originally limited to a single year (Zhou et al., 2021), we have expanded the weather dataset to encompass a decade of detailed weather data from 2014 to 2023, sourced consistently from the same weather station. The Weather-Captioned dataset now includes weather forecasting reports from a publicly accessible service, detailing the climate conditions in Jena, Germany, where the weather station is approximately located. These reports, updated every **six hours and daily**, cover a range of parameters including weather conditions, temperature, humidity, wind speed, and direction, providing a comprehensive overview for our analysis. As shown in Tab.1, each caption sample contains 7 sentences, each focus on a specific aspect. We directly use the channel name as channel description.

Specifically, in this dataset, the models aim to predict fine-grained time series for 21 channels, each with unique patterns and distributions, at 10-minute intervals based on 7 brief textual cues provided every 6 hours. This granularity refinement applies not only to the temporal dimension but also to the channel dimension.

We divided this extensive dataset into two subsets for detailed analysis: Weather-Captioned-Medium, covering data from 2014 to 2018, and Weather-Captioned-Large, which includes the entire dataset spanning ten years. This dataset encompasses over 525,600 weather data records across 21 different channels, with more than 85,000 unique sentences. This vast corpus provides millions of potential training samples for captioned time series forecasting. Further information is detailed in the Appendix L.4.

Table 1: Example caption of the Weather-Captioned dataset.

| Topic | Example |
|---|---|
| Month & Time of the Day | It's the early morning of a day in January. |
| Overall Weather | The current weather is clear. |
| Weather Trend in next 6h | The weather is expected to remain clear. |
| Temperature Trend in next 6h | The temperature is showing a mild drop. |
| Wind Speed & Direction | There is Light Breeze from NNW. |
| Atmosphere Pressure Level | The atmospheric shows Average Pressure. |
| Humidity Level | The air is very humid. |

## 5.4 STEAM GAME DATASET

A key application of TGTSF is in sales forecasting and decision-making. To illustrate this, we compiled data on online player counts from several of the most popular games a leading online video game distribution platform. This dataset also includes records of all game updates and events. On a view of game developer, we consider these updates and events as **controlled events**. The time series exhibits a weekly periodicity, with player activity typically peaking on weekends. Announcements from game developers frequently trigger spikes in activity, reflecting player enthusiasm for new content. However, the varied management strategies of game developers and the diverse reactions of players to updates introduce significant distribution shifts, even when textual information is considered. This variability makes the Steam Dataset the most challenging and intricate dataset within the TGTSF framework. [1]

## 6 EXPERIMENTS

We evaluate the TGForecaster model across four datasets to demonstrate the feasibility of the TGTSF task on the proposed datasets. As discussed in Section 5, each dataset is uniquely designed to test the task from different perspectives.

**Baseline Models** TGForecaster is benchmarked against state-of-the-art (SOTA) methods including DLinear, FITS, PatchTST, iTransformer, and Time-LLM (Zeng et al., 2023; Xu et al., 2023; Nie et al., 2023; Liu et al., 2023; Jin et al., 2023). We specifically contrast it with the linear-based models

---

[1] Due to intellectual property constraints, we may not be able to directly release this dataset. Detailed instructions are available in the appendix for researchers interested in replicating or extending our data.

DLinear and FITS to illustrate how TGForecaster surpasses periodicity-focused models. Comparisons with PatchTST highlight how textual information can enhance forecasting performance, even when using the same time series encoder. Time-LLM "reprograms" the LLaMa2 (Touvron et al., 2023), a large language model, which gives it capability of understanding text information. We use it as a baseline of multi-modal model. Specific settings differences, if any, are noted accordingly.

## 6.1 EVALUATION ON TOY DATASET

**Experiment Settings** All of the models are following the same experimental setup with prediction length $H \in \{14, 28, 60, 120\}$ and LBW length $T = 60$.

**Statistical Results** Table 2 presents the performance comparison of various models on the toy dataset, with TGForecaster significantly outperforming all baseline models. Specifically, TGForecaster achieved an 80% improvement in Mean Squared Error (MSE) over the best performing transformer-based model, and a 96% increase over DLinear. These results underscore the considerable benefits of integrating textual guidance in TSF. Notably, a version of TGForecaster without news data, which lacks the ability to utilize auxiliary textual information, demonstrated substantially lower performance. This underscores the essential role of text in enhancing forecast accuracy.

Table 2: Forecasting result on toy and Electricity-Captioned dataset in MSE. The best result is highlighted in **bold** and the second best is highlighted in underline.

| Dataset | Pred. Len. | TGForecaster | TGForecaster w/o News | FITS | DLinear | PatchTST | iTransformer | TimeLLM |
|---------|-----------|--------------|-----------------------|------|---------|----------|--------------|---------|
| Toy | 14 | **0.003** | 0.006 | 0.282 | 0.151 | 0.006 | 0.136 | 0.231 |
| | 28 | **0.008** | 0.018 | 0.692 | 0.297 | 0.029 | 0.295 | 0.382 |
| | 60 | **0.020** | 0.052 | 0.909 | 0.442 | 0.075 | 0.494 | 0.551 |
| | 120 | **0.027** | 0.102 | 0.883 | 0.632 | 0.168 | 0.747 | 0.788 |

| | Pred. Len. | TGForecaster | TGForecaster lbw 120 | FITS | DLinear | PatchTST | iTransformer | TimeLLM |
|---------|-----------|--------------|----------------------|------|---------|----------|--------------|---------|
| Elec-c /Elec | 96 | **0.124** | 0.127 | 0.134 | 0.140 | 0.130 | 0.148 | 0.131 |
| | 192 | **0.144** | 0.146 | 0.149 | 0.153 | 0.149 | 0.162 | 0.152 |
| | 336 | **0.160** | 0.164 | 0.165 | 0.169 | 0.166 | 0.178 | **0.160** |
| | 720 | 0.193 | 0.200 | 0.203 | 0.204 | 0.210 | 0.225 | **0.192** |

**Case Study** Figure 3 illustrates a segment of the toy dataset where the frequency changes within the forecasting horizon. In this visualization, the PatchTST model maintains the frequency observed in the look-back window, indicating an inability to adapt to new frequencies. Similarly, DLinear displays a collapsed pattern. Notably, without news input, TGForecaster shows the same behavior of PatchTST, suggesting that it relies solely on its time series encoder in the absence of textual cues. The models without external data also highlights the decreasing amplitude of predictions, a tendency to revert to safer, average predictions when faced with uncertainty in the far future—an issue known as the 'average shortcut' phenomenon discussed in Section 2.1. Conversely, with the integration of external text information eliminating uncertainty, TGForecaster adeptly adapts to new frequencies at the appropriate moments, demonstrating the substantial benefits of incorporating textual data into the forecasting process.

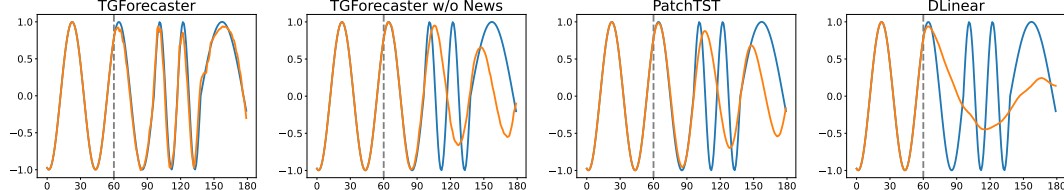

Figure 3: Visualization on the toy dataset. Look-back window is 60. Orange line for prediction and blue line for groundtruth. This segment shows two frequency change points in the forecasting horizon. With the help of external text information, TGForecaster can accurately adapt to new frequency at the correct position, however other conventional time series models fail to do so.

## 6.2 EVALUATION ON ELECTRICITY-CAPTIONED DATASET

**Experiment Settings** We follow the experiment settings in previous works as follows: forecasting horizon $H \in \{96, 192, 336, 720\}$, look back window length of 288. For fair comparison, we directly compare with the results report in the baseline original paper. And the TGForecaster is trained with

the captioned version. On this dataset, we also test the impact of a shorter look-back window on the TGForecaster. We will report accordingly in Tab. 2.

**Statistical Results** As depicted in Tab. 2, TGForecaster demonstrates SOTA performance on the Electricity dataset, particularly effective at shorter forecasting horizons using minimal textual cues. However, TimeLLM slightly outperforms TGForecaster over longer forecasting periods. Given TimeLLM's backbone is a large language model that also incorporates textual information, its edge may stem from its use of basic date and statistical data within its text inputs.

## 6.3 EVALUATION ON WEATHER-CAPTIONED DATASET

**Experiment Settings** We follow the experiment settings in previous works as follows: forecasting horizon $H \in \{96, 192, 336, 720\}$, look back window length of 360. We trained all other models on the Weather-Medium and -large dataset with their setting on the original weather dataset accordingly. And the TGForecaster is trained with the captioned version. We will report accordingly in Tab. 3.

Table 3: Forecasting result on Weather-Midium and -Large dataset in MSE. The best result is highlighted in bold and the second best is highlighted in underscore.

| Dataset | Pred. Len. | TGForecaster | FITS | DLinear | PatchTST | iTransformer | TimeLLM |
|---|---|---|---|---|---|---|---|
| Weather -Medium | 96 | **0.182** | 0.248 | 0.294 | 0.252 | 0.267 | 0.294 |
| | 192 | **0.205** | 0.297 | 0.340 | 0.304 | 0.327 | 0.342 |
| | 336 | **0.235** | 0.354 | 0.393 | 0.364 | 0.404 | 0.393 |
| | 720 | **0.281** | 0.430 | 0.456 | 0.439 | 0.495 | 0.461 |
| Weather -Large | 96 | **0.410** | 0.436 | 0.487 | 0.464 | 0.456 | - |
| | 192 | **0.438** | 0.524 | 0.568 | 0.567 | 0.578 | - |
| | 336 | **0.455** | 0.601 | 0.644 | 0.644 | 0.698 | - |
| | 720 | **0.497** | 0.692 | 0.725 | 0.745 | 0.832 | - |

**Statistical Results** As demonstrated in Tab. 3, TGForecaster significantly outperforms other models across both the Weather-Medium and Weather-Large datasets, substantiating the efficacy of incorporating external text information in addressing the information insufficiency inherent in TSF models. The results highlight that strategic integration of textual data can provide a more substantial performance boost than merely increasing the quantity of time series data. Further, we also report the channel-wise performance in Appendix D. We notice a groundbreaking performance boost of over **60%** on certain channels which were not predictable with historical time series data alone.

**Ablation Study** We conducted ablation studies to evaluate the impact of different embedding models and the integration of textual inputs on the performance of TGForecaster. We tested three embedding models: OpenAI Embedding (ope), paraphrase-MiniLM-L6 (Wang et al., 2020), and all-mpnet-base (Song et al., 2020). The results, presented in Tab. 4, indi-

Table 4: Ablation result on Weather-Captioned-Medium in MSE.

| Pred. Len. | Openai 512 | MiniLM | mpnet | MiniLM w/o Des. | MiniLM w/o News |
|---|---|---|---|---|---|
| 96 | **0.182** | 0.186 | 0.196 | 0.209 | 0.249 |
| 192 | **0.205** | 0.214 | 0.216 | 0.260 | 0.302 |
| 336 | 0.235 | **0.232** | 0.251 | 0.302 | 0.359 |
| 720 | 0.281 | **0.272** | 0.291 | 0.356 | 0.432 |

cate minimal performance differences between the embedding models. Notably, the removal of channel descriptions led to a significant performance decrease, underscoring the model's reliance on this feature for distinguishing between channels. Similarly, omitting news text resulted in performance dropping to levels comparable to the baseline PatchTST model, confirming that the improvements in forecasting accuracy are primarily driven by the inclusion of external textual information. This observation validates our hypothesis that textual data plays a crucial role in compensating for information deficiencies in traditional time series forecasting.

Our further ablation study demonstrates TGForecaster's ability to capture causal relationships between time series patterns and dynamic news. Detailed statistical result seen in Appendix G. When trained with correct and correlated dynamic news, the model effectively extracts these relationships, resulting in strong performance. However, when tested with random or misleading news, the model still tries to follow the causal relationship, producing poor results.

Conversely, when trained with random text, the model cannot establish causal correlations and reverts to PatchTST-level performance, relying only on the time series information. Even when correct news is provided during inference, the model, trained with bad text, disregards the text entirely, performing as if it had no text input. These findings collectively confirm TGForecaster's capability to extract causal relationships and its reliance on the quality of textual input.

**Visualization and Controllability Test** Fig. 4 visualizes three channels from the Weather-Caption-Medium dataset, full result see Appendix E. The first channel, atmospheric pressure (p), which is influenced by regional climate conditions, typically exhibits slowly changing trends that are challenging for TSF models to predict due to their subtle fluctuations without periodicity. However, with the integration of external information, TGForecaster accurately predicts these trends, whereas PatchTST tends to predict a constant average value, failing to capture the gradual changes.

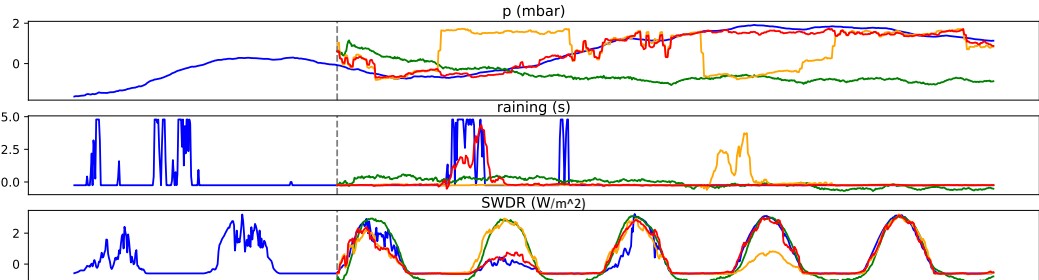

Figure 4: Visualization three channels on the 15000th test sample of Weather-Caption-Medium dataset. Blue line for ground truth, Red line for TGForecaster, Green line for PatchTST and Orange line for TGForecaster with swapping the news on the second and forth forecasting day.

The second channel, time of raining (measured in seconds per 10 minutes), lacks periodicity or clear causality and often appears binary. PatchTST typically predicts no rainfall, defaulting to an averaged shortcut. In contrast, TGForecaster adjusts its forecasts based on text prompts about upcoming rain but may miss predictions when the text weather report misaligns with actual events, as seen in the missed second rainfall period.

The third channel, SWDR (solar radiation), reflects the solar power reaching the ground. TGForecaster predicts SWDR shifts accurately by inferring inter-channel dependencies, even though solar radiation is not explicitly mentioned in the text, unlike PatchTST, which outputs basic waveform predictions.

Additionally, a controllability test swapping news inputs for the second and fourth days highlights TGForecaster's adaptability. It forecasts rain on the fourth day and clear conditions on the second, aligning predictions with the modified news data, demonstrating the model's responsiveness and effectiveness in text-guided forecasting.

### 6.4 Evaluation on Steam-100 Dataset

We evaluate the performance of the TGForecaster on our Steam-100 dataset, utilizing an input window of 60 days and an output horizon of 14 days. The findings indicate that the TGForecaster outperforms baseline models such as PatchTST, achieving a performance enhancement of over $12.6\%$. This superior performance is consistently observed across over $59.6\%$ of all games, ranking as the best among all evaluated methods. Given the significant stylistic variations among different game developers and the potential risk of temporal distribution shifts, traditional time series forecasting methods often struggle to capture the commonalities in temporal features. In contrast, the TGForecaster leverages textual information to significantly augment its predictive capabilities. We report the full result in the Appendix J.

## 7 Conclusions and Discussion

This paper addresses a critical roadblock in time series forecasting: information insufficiency. We introduced Text-Guided Time Series Forecasting (TGTSF), a new approach that integrates textual cues to enrich the models with external information and system knowledge. We developed and released four TGTSF datasets, each crafted to validate different aspects of the task and model. Our straightforward yet effective TGForecaster model demonstrates that textual guidance can significantly enhance time series modeling by mitigating the average predictions typically resulting from information scarcity.

While the TGForecaster effectively validates the TGTSF task, it does not fully comprehend the semantics of the text, such as extracting correlations among channels automatically. Future work will focus on advancing the model's semantic understanding and its ability to autonomously discern intricate relationships within the data.

## REPRODUCIBILITY & ETHIC STATEMENT

The code for TGForecaster and dataset samples are available at: `https://anonymous.4open.science/r/TGTSF_review-6E51`. For details, refer to Appendix A.

We comply with intellectual property agreements for all data sources. The weather report, as outlined in Appendix L.4, permits non-commercial use and will be released under the CC BY-NC-SA 4.0 license. Content generated by OpenAI API is free for general use, with no concerns regarding sensitive or illegal activity in our dataset. However, due to copyright constraints, the Steam dataset will not be published and is intended solely for evaluating our model within a household context (Appendix L.5).

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

## A  DATA AND CODE AVAILABILITY

The code for TGForecaster is avaliable at: `https://anonymous.4open.science/r/TGTSF_review-6E51`. Along with script for creating the toy and electricity dataset!

However, the Weather-Captioned dataset is to large for anonymous sharing. We upload a sample for inspection.

We will finally release all the time series, raw text and pre-embedded text embedding after the anonymous review period.

## B  PRELIMINARY WORKS

### B.1  TEXT EMBEDDING MODEL

Text embedding models have undergone significant advancements, providing efficient and semantically rich vector representations of textual information. Early transformer-based models like BERT (Devlin et al., 2019) encode sentences into embeddings by pretraining on masked language modeling tasks, enabling them to capture contextual semantics. However, BERT embeddings are not specifically optimized for tasks requiring fine-grained semantic similarity, prompting the development of more task-specific models.

MPNet (Song et al., 2020) and MiniLM (Wang et al., 2020) build upon BERT () by introducing novel architectural and pretraining strategies. MPNet combines masked language modeling with permuted sequence prediction, allowing for better contextual understanding and token dependencies. MiniLM, on the other hand, employs knowledge distillation to create smaller, faster models that retain high performance, making them ideal for resource-constrained applications.

OpenAI's embedding models (ope) represent another major step forward, leveraging large-scale proprietary transformer architectures. These embeddings are designed to excel in tasks like semantic search, classification, and similarity, offering generalizability and strong performance across a variety of applications. They also incorporate dimensional flexibility, allowing embeddings to be truncated or adjusted based on application needs, as seen with the Matryoshka embedding technique. This technique allows embeddings to maintain their semantic integrity even when their dimensions are reduced, offering scalability and adaptability.

A key property of text embeddings is their compatibility with similarity measures like cosine similarity. By projecting text into a shared semantic space, cosine similarity enables the computation of semantic closeness between embeddings, making it a foundational operation for tasks like clustering, retrieval, and alignment between modalities. This capability is crucial in applications requiring robust generalization across diverse textual expressions.

Together, these advancements have expanded the utility of text embeddings in various domains, including information retrieval, natural language understanding, and multimodal learning tasks. Our work builds on these innovations by leveraging pre-trained text embeddings for aligning textual semantics with time series patterns, ensuring robust causal modeling and efficient text-guided time series forecasting.

### B.2  TIME SERIES ANALYSIS WITH TEXT EMBEDDING

Adding more information to time series by incorporating heterogeneous information has been a long-studied topic, with several works opting to use text embeddings as input.

In the financial field, where time series are often more correlated to external information, several works (Sawhney et al., 2021; Liu et al., 2024b) have used text embeddings as external graph relationships to capture the correlations between keywords and stock descriptions, further influencing the ranking process in stock trading. More recently, a line of works (Liu et al., 2024a; Jia et al., 2024) has sought to enrich time series data by adding news text embeddings to the time series embeddings. However, these methods still face limitations in solving information insufficiency, as they do not incorporate causal information that could guide the model in predicting time series patterns driven by external events. Additionally, these works primarily use external text embeddings to expand the lookback window, without fully exploiting the underlying properties of the text embeddings.

To tackle these challenges, we introduce the Time-Series Guided Text Forecasting (TGTSF) model, which expands traditional time series forecasting by incorporating external textual data that offers causal insights. Unlike previous approaches that use text embeddings simply as supplementary information, TGTSF leverages the text to provide causal guidance, aligning textual data with time series patterns. Through the integration of TGCI, we can effectively extract channel-dynamic news correlations from the pre-trained text embeddings, enabling the model to adapt to the specific distributions of different time series channels. This allows the model to make more accurate predictions by incorporating both the semantic meaning of the text and its causal relationship with the time series data.

## C    ABOUT PREDICTABILITY OF TREND

In our study, we define "**trend**" as patterns that exhibit very low frequency while lacking periodicity within the observed time window, rather than simple exponential or linear patterns. For instance, the pressure channel in our Weather-Captioned dataset exemplifies this with its irregular low-frequency fluctuations, which appear to be random and non-periodic. Such randomness hampers the model's ability to learn stable patterns when relying solely on historical time series data.

However, these low-frequency patterns often correlate with external influences—for example, a drop in temperature due to cold air can significantly increase atmospheric pressure. By integrating this type of external information, our model is designed to discern causal relationships between such environmental factors and the observed low-frequency trends, thereby enhancing predictability.

For a practical illustration, please refer to the pressure (p-bar) channel in Fig. 4 and Fig. 5. This channel displays non-periodic fluctuations, which the traditional patchTST model even struggles produce a valid forecasting. In contrast, our TGTSF model, which incorporates external textual cues, successfully tracks these changes, demonstrating the effectiveness of including external information for predicting complex trends.

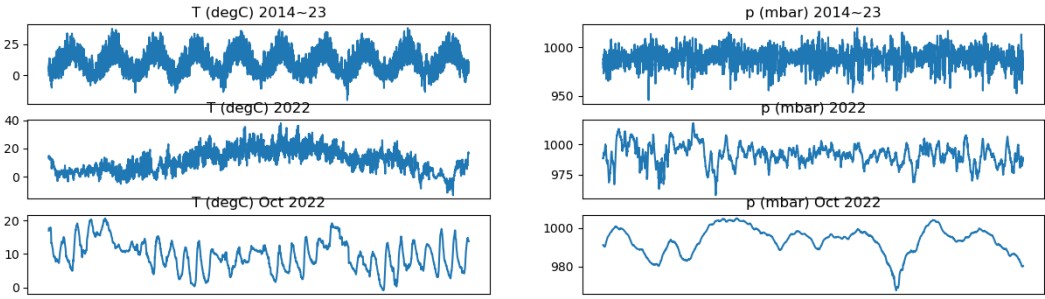

Figure 5: Visualization on two channels with different time scale. Temperature channel shows obvious periodicity both daily and annually. However the atmosphere pressure seems to be noise on large time scale but shows slowly random changing low-frequency "trend". Which makes it hard to be predicted without external information.

## D    CHANNEL-WISE PERFORMANCE ON WEATHER-CAPTIONED DATASET

The difficulty in predicting each channel varies, therefore, we present channel-wise performance in Table 5. The results demonstrate that TGForecaster, with the aid of external textual climate reports, significantly enhances forecasting accuracy across all channels. Notably, the model achieves over a 60% performance improvement in channels such as atmospheric pressure (p (mbar)), relative humidity (rh(%)), and vapor pressure deficit (VPdef (mbar)), which typically cannot be predicted reliably using historical time series data alone. The integration of external text cues has led to groundbreaking improvements in forecasting these parameters.

However, the wind velocity channel shows minimal variance in performance across all models, each achieving similar results with slight losses. This phenomenon is attributed to the presence of extreme values in this channel, which, after normalization, diminish the impact of more typical values on the overall gradient. Consequently, all models struggle to learn detailed patterns in this channel due to the reduced contribution to the global gradient.

Another noteworthy observation is that while TGForecaster is capable of predicting rainfall—unlike models that default to predicting near-zero averages—the performance improvement in these channels is modest. This is because rainfall is relatively scarce in this dataset, leading to large losses when rain is inaccurately predicted at the wrong times. Conversely, predicting the average value results in a smaller overall loss. This tendency explains why other models often opt for the average, avoiding the complex task of learning rainfall patterns. Nevertheless, accurate rainfall forecasting remains crucial in meteorological applications, underscoring our commitment to enhancing predictive accuracy in this area. The same principle also applies to other channels.

Table 5: Channel wise performance on Weather-medium dataset in MSE. The best is highlighted in bold and the second best is highlighted in underline.

| Channel | TGForecaster | FITS | DLinear | PatchTST | iTransformer | IMP. |
|---|---|---|---|---|---|---|
| p (mbar) | **0.1365** | 0.8637 | 0.8238 | 0.9301 | 1.0320 | **83.43%** |
| T (degC) | **0.1889** | 0.2924 | 0.3329 | 0.2964 | 0.3233 | 35.40% |
| Tpot (K) | **0.1829** | 0.3163 | 0.3525 | 0.3225 | 0.3533 | 42.18% |
| Tdew (degC) | **0.3467** | 0.4043 | 0.4085 | 0.4082 | 0.4258 | 14.25% |
| rh (%) | **0.2479** | 0.6541 | 0.6788 | 0.6997 | 0.8185 | **62.10%** |
| VPmax (mbar) | **0.2369** | 0.3500 | 0.3984 | 0.3521 | 0.4086 | 32.31% |
| VPact (mbar) | **0.2998** | 0.3404 | 0.3534 | 0.3515 | 0.3845 | 11.93% |
| VPdef (mbar) | **0.2835** | 0.6384 | 0.6968 | 0.6744 | 0.8038 | **55.59%** |
| sh (g/kg) | **0.2995** | 0.3434 | 0.3562 | 0.3557 | 0.3896 | 12.78% |
| H2OC (mmol/mol) | **0.2996** | 0.3434 | 0.3562 | 0.3556 | 0.3894 | 12.75% |
| rho (g/m³) | **0.1926** | 0.3909 | 0.4119 | 0.4182 | 0.4535 | **50.73%** |
| wv (m/s) | **0.0002** | **0.0002** | **0.0002** | **0.0002** | 0.0003 | 0.00% |
| max. wv (m/s) | 0.0004 | 0.0005 | **0.0004** | 0.0005 | 0.0006 | 0.00% |
| wd (deg) | **0.7270** | 1.1605 | 1.1295 | 1.1344 | 1.2735 | 35.64% |
| rain (mm) | **0.6824** | 0.6905 | 0.7167 | 0.6891 | 0.6998 | 0.97% |
| raining (s) | **0.7900** | 0.8735 | 0.9379 | 0.8591 | 0.9942 | 8.04% |
| SWDR (W/m²) | **0.1828** | 0.3084 | 0.3856 | 0.2967 | 0.3776 | 38.39% |
| PAR (umol/m²/s) | **0.1773** | 0.2840 | 0.3588 | 0.2704 | 0.3473 | 34.43% |
| max. PAR (umol/m²/s) | **0.1975** | 0.2599 | 0.3195 | 0.2632 | 0.3226 | 24.01% |
| Tlog (degC) | **0.1774** | 0.2802 | 0.3260 | 0.2806 | 0.3290 | 36.69% |
| CO2 (ppm) | **0.2600** | 0.2716 | 0.2812 | 0.2618 | 0.2760 | 0.69% |
| Avg. Loss | **0.2814** | 0.4317 | 0.4583 | 0.4391 | 0.4954 | 34.82% |

## E  PERFORMANCE VISUALIZATION ON WEATHER-CAPTIONED DATASET

We provide the full visualization as Fig. 6. The TGForecaster shows great performance across all the channels. Even very hard ones such as Wind dir. It can also model the time series that totally independent with the weather such as the CO2 channel.

## F  WEATHER RESULTS W. W/O. RIN

We compared the performance of models with and without Reversible Instance Normalization (RIN) on the weather-captioned-medium dataset, focusing on a 720-hour forecasting horizon. The model with RIN enabled achieved an MSE of 0.3428, whereas the model without RIN achieved a lower MSE of 0.2814. Results visualized in Fig. 7 show that the RIN-enabled model exhibits significant biases in many channels, particularly those with gradual trend shifts. This occurs because RIN removes the bias term from all instances, leaving the model unable to recognize relative bias and trend values. For instance, with RIN, temperature patterns in winter and summer are treated similarly, ignoring the typically higher and more variable temperatures in summer. Additionally, we noted pronounced shifting behavior coinciding with changes in captions, suggesting that the absence of bias information leads the model to over-rely on textual prompts, compensating for the missing data.

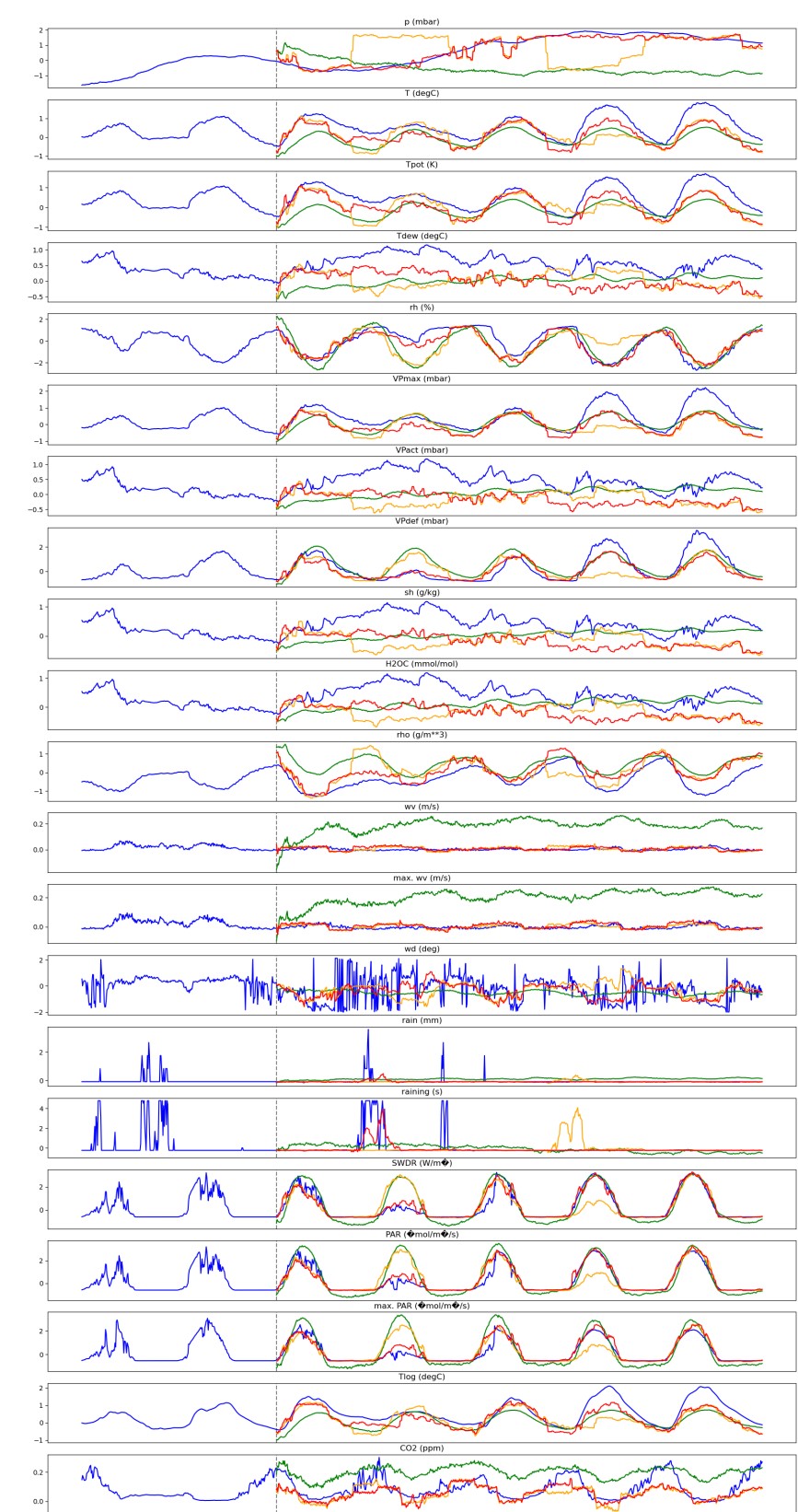

Figure 6: Full visualization all channels on the 15000th test sample of Weather-Caption-Medium dataset. Blue line for ground truth, Red line for TGForecaster, Green line for PatchTST and Orange line for TGForecaster with swapping the news on the second and forth forecasting day.

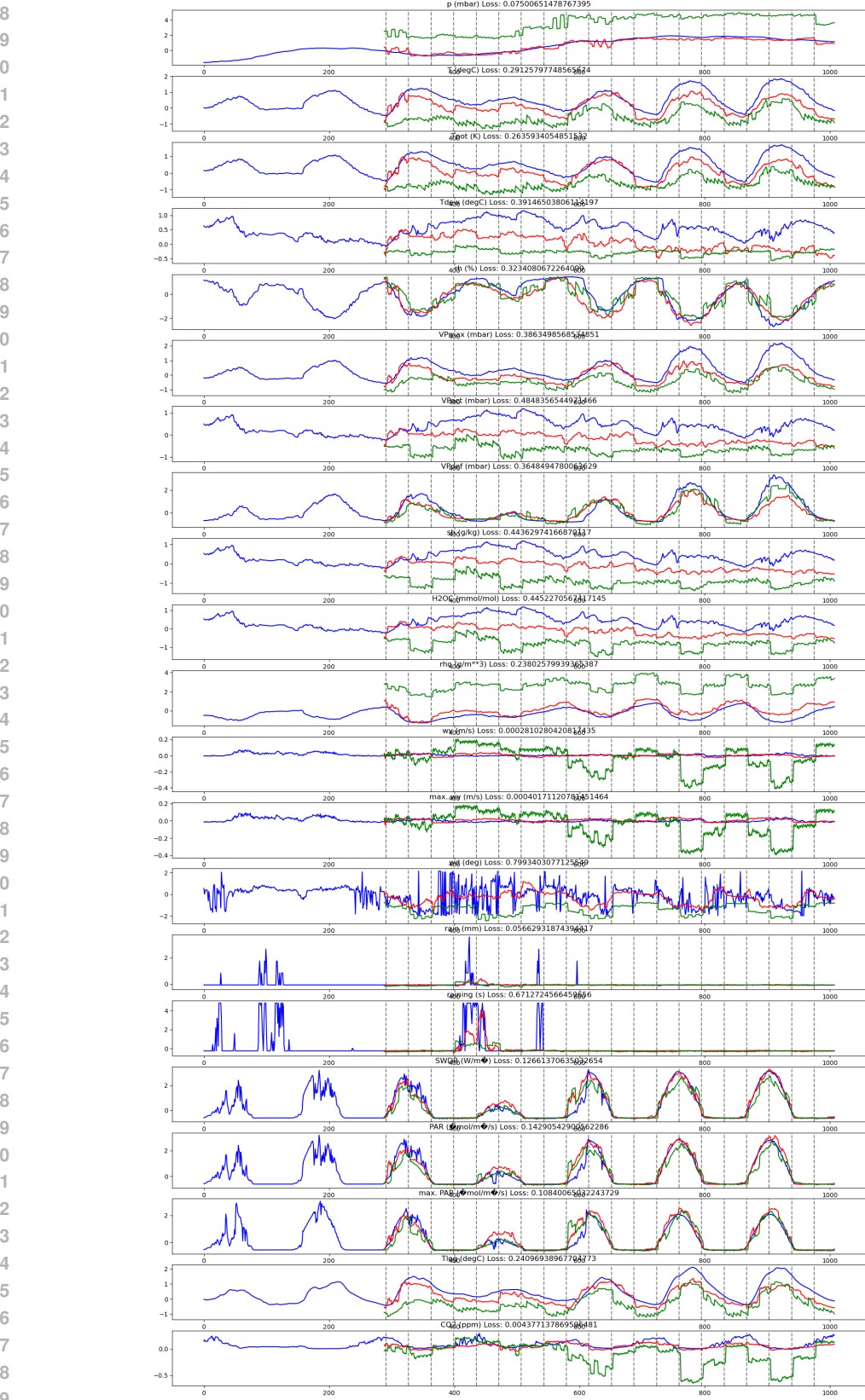

Figure 7: Full visualization all channels on the 15000th test sample of Weather-Caption-Medium dataset. Blue line for ground truth, Red line for TGForecaster without RIN, Green line for TGForecaster with RIN enabled.

# G    ABLATION STUDY ON CAUSAL RELATIONSHIP EXTRACTION

TGForecaster is designed to learn causal relationships between events described in text and their corresponding time series patterns. While not explicitly an alignment model, it effectively aligns the semantic meaning of text with the time series data it impacts. The model generates time series patterns guided by the textual information, and its performance varies based on the quality of the text input:

1. **Training with Meaningful and Relevant Text:**
   - **Inference with Similar Text:** Produces strong results by accurately extracting causal relationships between events in the text and time series patterns.
2. **Training with Zero/Random Text:**
   - **Inference with Any Text:** Produces results equivalent to PatchTST, as no additional information is present in the text. The model relies solely on the time series data, ignoring the random text.
3. **Training with Meaningful Text, Inference with Incorrect Text:**
   - **Inference with Incorrect Text:** Results are poor, as the model relies on the misleading text input and generates patterns based on incorrect or irrelevant information.

We detail the TGForecaster performance under different text conditions in the following table:

Table 6: TGForecaster performance under different training and testing conditions. We report the result of forecasting horizon 96 on weather-captioned-medium dataset using MiniLM embedding. Note that the numerical result are not final and may subject to change in the final version.

| | | Train with | | |
|---|---|---|---|---|
| | | Good | Zero | Random |
| Test with | Good | 0.186 (captures causal relationships) | 0.249 (corrupted random patterns) | 0.251 (similar to PatchTST) |
| | Zero | 0.724 (corrupted repetive patterns) | 0.249 (similar to PatchTST) | 0.254 (similar to PatchTST) |
| | Random | 0.615 (corrupted random patterns) | 0.249 (similar to PatchTST) | 0.250 (similar to PatchTST) |

The results of the ablation study provide strong evidence that TGForecaster relies on capturing causal relationships between time series patterns and dynamic news, rather than simply treating text as auxiliary input. When trained with meaningful and correlated news, the model demonstrates its ability to effectively extract these relationships, yielding strong predictive performance. This highlights TGForecaster's capacity to align the semantic meaning of text with time series patterns in a causally meaningful way.

On the other hand, when trained with good text but tested with random or misleading text, the model produces poor predictions because it continues to rely on the input text, even when it is inaccurate or irrelevant. This further underscores the model's dependence on the quality of the textual input rather than merely defaulting to learned time series patterns.

Interestingly, when trained with bad or random text, TGForecaster fails to establish causal relationships and instead reverts to PatchTST-level performance, indicating it falls back to relying solely on time series data. Furthermore, when subsequently tested with good text, the model trained on bad text still ignores the input entirely, suggesting it stops depending on textual input when the training data lacks meaningful causal relationships.

These results collectively demonstrate that TGForecaster's strength lies in its ability to extract and leverage causal relationships between text and time series data. The model's performance is tightly coupled with the quality and relevance of the textual input, validating the centrality of causal alignment in its design and functionality.

These outcomes demonstrate that TGTSF effectively achieves alignment in the "event" space, linking events described in the text to the corresponding time series patterns.

## H   ATTENTION MAP VISUALIZATION ON WEATHER-CAPTIONED

We further visualize two cross-attention blocks to further investigate the TGForecaster. You are strongly advised to check the Tab. 1, Appendix L.4.6 and Fig. 6 while reading this part.

Figure 8 illustrates the attention map of the "text-guided channel independent" cross-attention block in the text encoder across three layers. In the first layer, attention is predominantly focused on the first sentence, which specifies the month and time. This sentence is crucial as it provides temporal context that significantly impacts the prediction of both daily and annual periodicity. While other sentences receive moderate attention, the sixth sentence, which describes atmospheric pressure as detailed in Table 1, consistently receives no attention across all channels.

In the second layer, however, there is a notable shift in attention dynamics. All channels, particularly channel 0, show intense focus on the sixth sentence. According to the channel definitions in Appendix L.4.6, channel 0 directly corresponds to atmospheric pressure. Channels 10 and 20, which are related to air density and CO2 concentration respectively—factors closely associated with pressure—also display relatively high attention scores. This suggests that the TGForecaster is capable of discerning the underlying relationships among the channels.

The separation of attention focus between the first and second layers suggests that the influence of atmospheric pressure on the model's predictions is independent of time. In the third layer, a diversity of attention patterns emerges; channel 0 focuses exclusively on the sixth sentence, while other channels predominantly attend to the first sentence.

Since we take the output of previous layer as query and input news embeddings as key and value, the information lies in the news are progressively added to the channel embeddings. Thus, the model can focus on different perspective in separate cross attention layers.

Figure 9 presents the attention map of the modality mixer layer cross attention block in the weather-captioned dataset. The map, averaged across three cross attention layers, illustrates distinct patterns of attention for each channel. This diversity underscores the TGForecaster's ability to adaptively extract time series embeddings tailored to the unique distribution characteristics of each channel, facilitated by textual inputs.

Notably, the channels for SWDR, PAR, and max.PAR display clear periodic patterns in their attention maps, aligning with observations from waveform visualizations. These patterns suggest that the TGForecaster effectively captures and utilizes periodic information from these environmental variables.

Furthermore, the channels labeled rain and raining show a particularly interesting behavior; they assign significantly higher attention scores to the exact time periods of rainfall within the look-back window. This behavior indicates that the TGForecaster is adept at identifying and prioritizing crucial temporal events specific to each channel, further enhancing its forecasting accuracy by focusing on relevant patterns where needed. This level of detail in attention allocation demonstrates the model's capability to integrate contextual cues from textual data and further guide the time series forecasting.

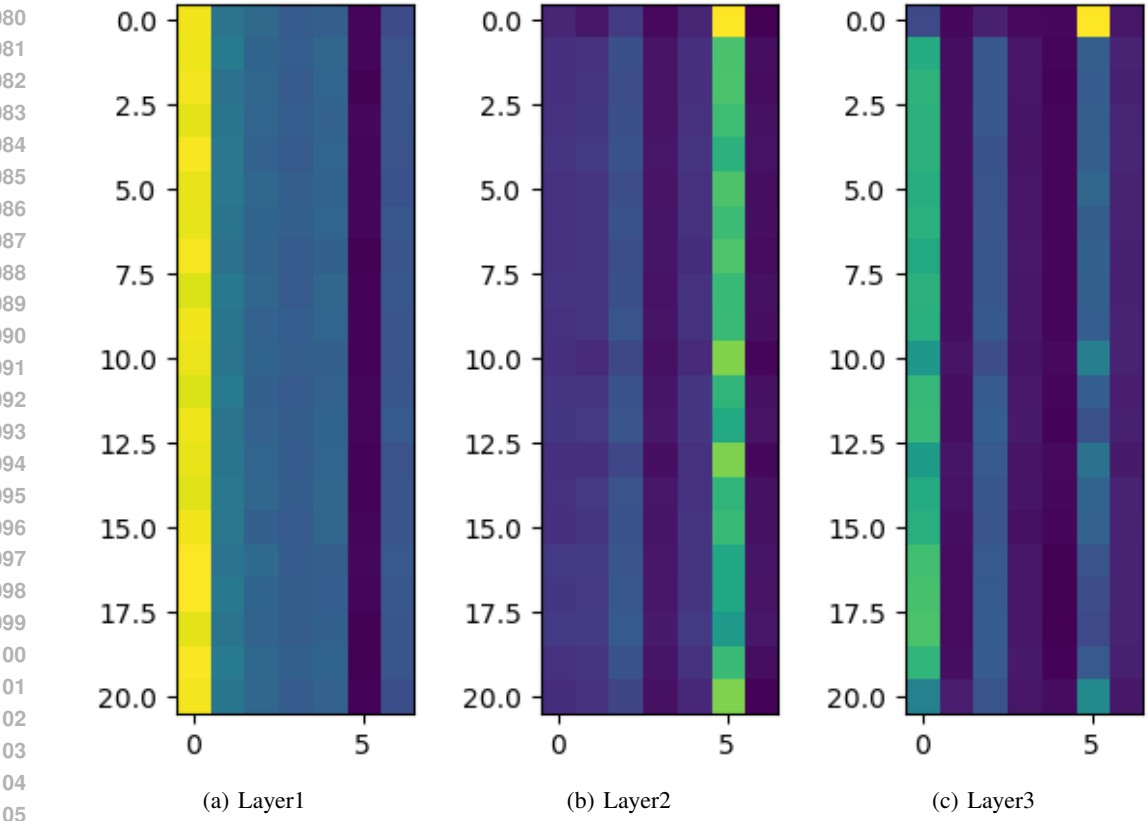

(a) Layer1          (b) Layer2          (c) Layer3

Figure 8: Attention map of the "text guided channel independent" cross attention block on weather-captioned dataset, on the 15000th test sample of Weather-Caption-Medium dataset. We use three cross attention block. The vertical axis stand for channels and horizon stand for the 7 sentences of the weather report summary.

## I    COMPARISON WITH MORE BASELINES ON ELECTRICITY-CAPTIONED

We further compare with more baselines on Electricity-Captioned, including Autoformer, Fedformer, Informer, FiLM and TimesNet (Wu et al., 2021; Zhou et al., 2022a; 2021; 2022b; Wu et al., 2023). TGForecaster shows dominant superior performance across these baselines, as shown in Tab. 7.

Table 7: The comparison on Electricity dataset with other baselines. Best is marked in bold and the second best is marked in underline.

|  | Pred. Len. | TGForecaster | TGForecaster_120 | Autoformer | Fedformer | Informer | FiLM | TimesNet |
|---|---|---|---|---|---|---|---|---|
|  | 96 | **0.124** | 0.127 | 0.201 | 0.188 | 0.274 | 0.154 | 0.168 |
|  | 192 | **0.144** | 0.146 | 0.222 | 0.197 | 0.296 | 0.164 | 0.184 |
| Elec | 336 | **0.16** | 0.164 | 0.231 | 0.212 | 0.3 | 0.188 | 0.198 |
|  | 720 | **0.193** | 0.200 | 0.254 | 0.244 | 0.373 | 0.236 | 0.22 |

## J    FULL RESULT ON STEAM-100 DATASET

We show the comparison on Steam-100 Dataset in Tab. 8, and Tab, 9 with PatchTST. We use de-normed MAE as metric since the base volume of players of each game varies drastically, using normed metrics can lead to unfair comparison. The pretrain indicate that the model is jointly trained on all the games and each game is labeled by the channel discription. The Gnorm indicate that we apply the global normalization to preserve the player variation mentioned before. But it seems bring limited boost.

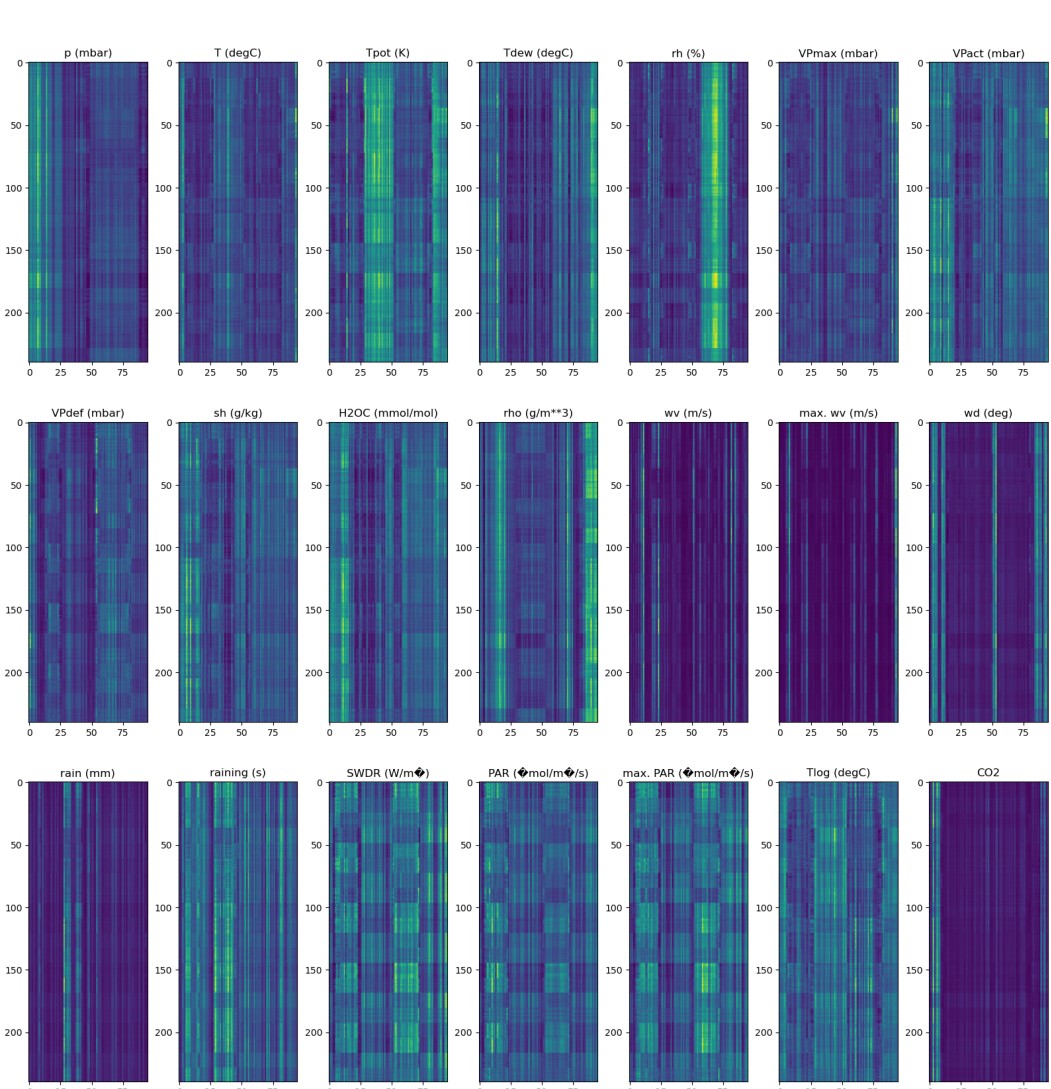

Figure 9: Attention map of the modality mixer layer cross attention block on weather-captioned dataset, on the 15000th test sample of Weather-Caption-Medium dataset. The attention map is averaged across three cross attention layers. We plot the attention map for each channel. The vertical axis stand for output time series patches and the horizon stand for the input time series patches embedding from PatchTST backbone.

Table 8: Full result on Steam-100 dataset in de-normalized MAE. The best result is shown in green shaded bold font. The ones with performance boost over 10% is marked in red.

| game_id | TGTSF_pretrain | TGTSF_pretrain_Gnorm | TGTSF | PatchTST | IMP/% |
|---|---|---|---|---|---|
| 10 | 781.2257 | **724.5196** | 849.8968506 | 804.757019 | 0.099704 |
| 240 | **372.887** | 374.03445 | 400.0899353 | 421.2349243 | 0.114777 |
| 440 | **10216.276** | 10816.979 | 10694.47852 | 10828.37891 | 0.056528 |
| 550 | **4110.673** | 4437.846 | 4336.002441 | 4587.186035 | 0.103879 |
| 570 | **30179.111** | 30914.955 | 31916.02344 | 32031.38477 | 0.057827 |
| 620 | **674.1199** | 789.6992 | 799.2894897 | 710.8320313 | 0.051647 |
| 730 | 51687.87 | **50937.348** | 52638.79297 | 51069.08984 | 0.00258 |
| 3590 | **687.4657** | 775.8215 | 734.3128662 | 743.4817505 | 0.075343 |
| 39210 | **3310.422** | 3719.8003 | 3388.064453 | 3420.974609 | 0.032316 |
| 105600 | **4415.245** | 4899.586 | 4464.801758 | 4513.034668 | 0.021668 |
| 107410 | 1597.8743 | 1548.8544 | 1411.974243 | **1355.757813** | 0 |
| 214950 | 331.97876 | 350.3743 | 328.4424744 | **324.0109863** | 0 |
| 218620 | **5576.539** | 5714.511 | 5650.027832 | 5818.570313 | 0.041596 |
| 221100 | **2574.3164** | 2713.0063 | 3260.266113 | 2742.404053 | 0.061292 |
| 222880 | **50.29491** | 138.43388 | 61.51412582 | 59.99531174 | 0.161686 |
| 227300 | **3224.443** | 3356.0312 | 3418.429199 | 3388.108398 | 0.048306 |
| 230410 | **5307.4634** | 5733.2676 | 5856.409668 | 6040.648926 | 0.121375 |
| 231430 | 441.79767 | **377.2723** | 581.7993774 | 713.0888062 | 0.470932 |
| 232050 | **5.8684874** | 140.72949 | 6.452753067 | 6.638870239 | 0.116041 |
| 236390 | **4528.812** | 4847.2886 | 4787.857422 | 4680.506348 | 0.03241 |
| 236850 | 1197.9114 | 1308.7864 | **1169.570313** | 1191.723145 | 0.018589 |
| 242760 | **5888.873** | 6968.5566 | 6002.225586 | 6160.327148 | 0.044065 |
| 244210 | **657.4606** | 672.32324 | 676.5147095 | 658.0761108 | 0.000935 |
| 250900 | 895.2101 | **886.4052** | 1012.618652 | 892.572937 | 0.00691 |
| 251570 | 4304.9175 | 4886.079 | **4088.169922** | 4352.344727 | 0.060697 |
| 252950 | 1971.4385 | **1928.533** | 1971.546509 | 2054.135742 | 0.061146 |
| 255710 | 2154.8572 | 2082.0603 | 2231.175781 | **2078.98584** | 0 |
| 270880 | 789.74634 | 748.85596 | **746.2268677** | 760.31073 | 0.018524 |
| 271590 | **9292.364** | 9546.938 | 10758.82422 | 9438.995117 | 0.015535 |
| 275850 | **2671.1484** | 3121.9731 | 3179.639404 | 3118.741699 | 0.143517 |
| 281990 | **2094.3948** | 2404.5767 | 3269.309326 | 2315.452881 | 0.095471 |
| 284160 | 929.92413 | **903.6123** | 956.4987183 | 908.8114014 | 0.005721 |
| 289070 | 4861.033 | 5243.972 | 4706.715332 | **4633.394531** | 0 |
| 291550 | 1339.3384 | 1323.4893 | **1290.662231** | 1324.343018 | 0.025432 |
| 292030 | **3181.2307** | 3723.4417 | 3607.125732 | 3500.928223 | 0.091318 |
| 294100 | 2123.2292 | 2150.6 | **1990.630981** | 2074.705322 | 0.040524 |
| 304930 | 5698.3926 | 5597.64 | **5430.058105** | 5824.242188 | 0.06768 |
| 306130 | **1727.7567** | 1927.9446 | 1787.971802 | 1765.812744 | 0.021552 |
| 322170 | 987.1338 | **878.4059** | 902.7176514 | 910.0273438 | 0.034748 |
| 322330 | **4585.129** | 5155.708 | 4900.762207 | 4916.681152 | 0.067434 |
| 346110 | **6389.454** | 7178.1084 | 7004.43457 | 6871.126953 | 0.070101 |
| 359550 | 5251.984 | 5247.6694 | 5184.080566 | **5156.916016** | 0 |
| 364360 | **78.73614** | 182.35359 | 89.08701324 | 88.43521118 | 0.109674 |
| 365590 | **158.19469** | 229.93842 | 173.8761902 | 180.6734314 | 0.124416 |
| 374320 | **557.42993** | 630.0981 | 659.1234131 | 655.4638672 | 0.149564 |
| 377160 | 1733.9108 | **1486.6573** | 1814.430298 | 1725.577515 | 0.138458 |
| 381210 | **5498.251** | 5568.4336 | 5691.748047 | 5964.625488 | 0.07819 |
| 386360 | 1135.4182 | 1231.7946 | 1126.588989 | **1125.822021** | 0 |
| 394360 | 2788.8633 | **2674.6377** | 2861.023682 | 2725.428223 | 0.018636 |
| 413150 | 3061.9893 | 3204.8018 | 2831.16748 | **2713.414551** | 0 |
| 427520 | 923.96985 | **772.3062** | 803.0150146 | 775.4301758 | 0.004029 |
| 457140 | 1159.0262 | 1217.2543 | 1133.615601 | **1107.060547** | 0 |
| 489830 | 2071.2698 | 2067.1377 | 1855.288574 | **1809.03479** | 0 |
| 493520 | **270.12488** | 337.2064 | 311.7146606 | 314.7081604 | 0.141665 |
| 513710 | **1662.712** | 1732.691 | 2458.639648 | 2096.899658 | 0.207062 |
| 526870 | 1714.113 | **1704.2557** | 2002.178223 | 2031.111084 | 0.160924 |

Table 9: Cont. Full result on Steam-100 dataset in de-normalized MAE. The best result is shown in green shaded bold font. The ones with performance boost over 10% is marked in red.

| game_id | TGTSF_pretrain | TGTSF_pretrain _Gnorm | TGTSF | PatchTST | IMP/% |
|---|---|---|---|---|---|
| 529340 | **2037.8783** | 3383.1628 | 4038.390381 | 3701.234863 | 0.449406 |
| 548430 | **3215.861** | 3539.4717 | 3559.275391 | 3622.380615 | 0.112224 |
| 552500 | **2653.9133** | 2886.299 | 3560.82251 | 4062.790527 | 0.346776 |
| 552990 | 5025.465 | 5687.0366 | **3085.568848** | 5004.588379 | 0.383452 |
| 578080 | 21942.736 | **20576.87** | 24925.38086 | 21725.19727 | 0.052857 |
| 582010 | **2820.5447** | 3062.556 | 3161.88623 | 3088.219727 | 0.086676 |
| 582660 | **1543.2185** | 1686.1365 | 1597.595703 | 1613.270874 | 0.043423 |
| 646570 | **1219.2263** | 1304.4564 | 1372.947632 | 1420.136963 | 0.141473 |
| 648800 | **1754.793** | 2505.1694 | 2167.084717 | 2164.018311 | 0.189104 |
| 739630 | **3801.0945** | 4393.0825 | 4291.587891 | 4325.070313 | 0.121149 |
| 761890 | 1032.8406 | 1291.7933 | 1046.670288 | **1021.805847** | 0 |
| 814380 | **1362.8702** | 1614.8129 | 1658.933594 | 1566.417725 | 0.129945 |
| 892970 | 3217.5664 | **2532.8167** | 3313.418701 | 3676.412598 | 0.311063 |
| 960090 | **1899.8359** | 2076.6377 | 2291.049316 | 2292.431396 | 0.171257 |
| 1085660 | **16838.436** | 18822.541 | 18422.99219 | 18631.20313 | 0.096224 |
| 1091500 | **13698.44** | 14906.672 | 15611.44824 | 16007.56543 | 0.144252 |
| 1172470 | **33071.402** | 39898.113 | 37495.97266 | 37135.38672 | 0.109437 |
| 1172620 | **2595.5396** | 3012.9434 | 2990.058838 | 3043.287109 | 0.147126 |
| 1222670 | 3100.3125 | **2550.8154** | 3211.132813 | 3179.268799 | 0.197672 |
| 1238810 | **1900.4874** | 1945.948 | 2186.150391 | 2113.537598 | 0.100803 |
| 1238840 | **1193.3369** | 1223.7898 | 1373.025757 | 1397.660156 | 0.14619 |
| 1293830 | 1417.9685 | 1575.0452 | **1349.839478** | 1393.606567 | 0.031406 |
| 1326470 | **1735.4362** | 1926.2095 | 2924.927979 | 2730.827881 | 0.364502 |
| 1361210 | **4610.036** | 4732.349 | 9080.219727 | 6853.158203 | 0.327312 |
| 1454400 | 809.03754 | **769.7717** | 941.6995239 | 1514.508057 | 0.491735 |
| 1623660 | **576.48615** | 741.7087 | 850.6308594 | 978.7790527 | 0.411015 |
| 1665460 | 1282.9064 | **1174.2739** | 1324.958374 | 1345.970093 | 0.127563 |
| 1677740 | 1610.0262 | 1459.8435 | **1413.402588** | 1450.460693 | 0.025549 |
| 1811260 | **4966.4424** | 9192.259 | 8108.043457 | 7732.84668 | 0.357747 |
| 1868140 | 1495.137 | **1424.7719** | 12159.14551 | 9878.760742 | 0.855774 |
| 1919590 | **1274.8295** | 2378.603 | 6667.467773 | 1967.391602 | 0.35202 |
| 1938090 | **10918.149** | 10952.329 | 14469.45508 | 14213.95605 | 0.231871 |
| 1948980 | **534.38995** | 725.96063 | 1041.738037 | 1054.809937 | 0.493378 |
| Best_count | 53 | 17 | 9 | 10 | 0.126324 |

Table 10: The mean and std of TGForecaster on the three dataset in metrics of MSE.

| Datasets | TGForecaster | FITS |
|---|---|---|
| Toy | 0.027±0.001 | 0.883±0.000 |
| Electricity | 0.193±0.004 | 0.203±0.001 |
| Weather-Medium | 0.281±0.008 | 0.430±0.011 |

## K  Error Bar & Critical Difference Diagram

We run the experiments on Toy and Electricity for five times with different randomly chosen random seeds. And Weather-Medium for three times because of the large amount of data can result in very long training time on our devices. We report the mean and standard deviation as follows with comparison with FITS, the most stable model.

As Tab. 10 indicate, TGForecaster shows stable performance across the benchmark. Even with extreme condition, it still maintains superior performance. It worth note that, we thought the relative large variance on weather dataset is caused by the different combination of the text description. But the FITS also shows large variance on this dataset which indicate it is hard to converge on this dataset.

We generate the critical difference plot on our result of four datasets (toy, Electricity, Weather-Medium, Weather-Large) with the default alpha as 0.05 as shown in Fig. 10. TGForecaster's placement at the top of the critical difference plot, without intersecting with other lines, demonstrates its consistent and superior performance in terms of MSE compared to the other models. It indicates that with the help of external textual information, TGForecaster can handle complicated datasets.

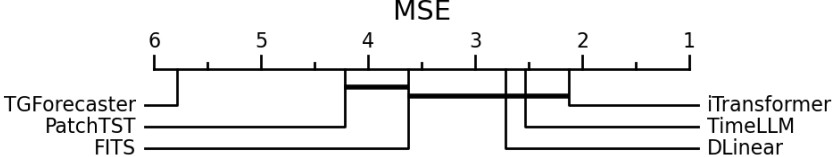

Figure 10: The Critical Difference Plot on the TGForecaster and other baselines with alpha=0.05.

## L  TGTSF Benchmark Datasets

We designed the TGTSF benchmark to include four datasets of varying complexity, each tailored to evaluate specific aspects of model performance. Together, these datasets form a progression from simple, interpretable scenarios to challenging, real-world applications, providing a comprehensive evaluation framework for text-guided time series forecasting models.

*1. Toy Dataset* The Toy dataset is intentionally designed with simple and straightforward patterns, making it easy to analyze and interpret. However, the dataset includes sudden changes in patterns that are impossible to predict without text guidance. This ensures the model's ability to adhere to textual cues is effectively tested in a controlled environment. It serves as a foundation for validating whether the model can extract and use textual guidance to forecast time series.

*2. Electricity Dataset* The Electricity dataset introduces real-world data with common textual features like day of the week or public holidays. While the textual information is relatively simple, it tests the model's ability to utilize such structured cues for forecasting. Additionally, as a widely-used off-the-shelf dataset, it allows for easy comparison with existing methods, providing a baseline for evaluating TGTSF's performance.

*3. Weather-Captioned Dataset* The Weather-captioned dataset represents a semi-controlled environment designed to rigorously test the model's ability to learn causal relationships between text and time series patterns. It also evaluates the model's text-guided channel independence and generalizability. By simulating a scenario where text and time series data are strongly correlated, this dataset bridges the gap between controlled tests and more complex real-world challenges.

*4. Steam Dataset* The Steam dataset is a fully real-world dataset that tests TGTSF in a practical industrial context. Its patterns are noisy and random, making it highly challenging. This dataset showcases TGTSF's ability to perform well in realistic scenarios. Although we cannot release the

dataset due to intellectual property restrictions, we will provide detailed instructions for replicating it in the code repository.

**Comprehensive Benchmark Objectives**

The TGTSF benchmark is designed to address multiple objectives:

- **Interpretability and Validation:** The simpler Toy and Electricity datasets help researchers validate their models and understand their behavior in controlled environments.
- **Performance Testing in Complex Scenarios:** The Weather-captioned and Steam datasets challenge the models in semi-controlled and real-world settings, ensuring they are robust and capable of handling practical applications.

This benchmark is not merely a ranking tool but a framework to help researchers analyze and improve their models' behaviors across varying levels of complexity. We see this as a starting point for the community and hope it will inspire researchers to contribute additional datasets, further expanding and enriching the TGTSF benchmark for future advancements in this field.

## L.1 METADATA FOR DATASETS

We show the metadata for TGTSF Datasets in Table 11.

Table 11: Datasets Metadata

| Dataset | Length | Time span | TS Sampling Rate | # of Channels | # of Dynamic News each step | Textual update rate | Notes |
|---|---|---|---|---|---|---|---|
| Toy | 300,000 | N/A | N/A | 1 | 1∼3 | Every Step | Sinusoidal wave with a single channel |
| Electricity-Captioned | 26,304 | 2011-01-01 to 2015-12-31 | 1 hour | 321 | 1∼3 | Daily | Just the Electricity Dataset |
| Weather-Captioned | 525,600 | 2014-01-01 to 2023-12-31 | 10 minutes | 21 | 7 | Every 6 hours | Weather data with 21 channels. Three set of textual cues for combination. |
| Steam | Varies | 2005 to 2024 | 1 Day | 1 (each game) | Varies | Varies | 100 popular games on Steam. Each game has historical data from its prelaunch to 2024. |

## L.2 TOY DATASET DETAILS

We directly generate this dataset with sinusoidal wave that randomly changes frequency. Before each changing point, we add 10 captions as 'Channel 1 will change to frequency x in y timesteps.' After each changing point, we add 5 captions as 'Channel 1 will keep steady with frequency of x.' In other timesteps, we caption it as 'The waveform will go steady.'

We will publish this dataset with CC BY-NC-SA 4.0 licence.

## L.3 ELECTRICITY-CAPTION DETAILS

We caption the day of week with the given time stamp. But we somehow find the original time stamp is incorrect. Instead of the year of 2016, it should be collected in year 2012. Without knowing the exact location of this building, we cannot identify the specific public holiday. We then uses channel 319, which shows obvious patterns of workday and holiday as indicator, when the average value lower than a specific value, we caption it with public holiday.

We will publish this dataset with CC BY-NC-SA 4.0 licence.

## L.4 WEATHER-CAPTION DETAILS

### L.4.1 DATA SOURCE

In creating a TGTSF dataset, it is advisable to avoid directly generating the description out of the forecasting horizon time series pattern as news messages, as this could lead to information leakage. News messages should instead contain relevant, known information from other sources. Thus, we get the weather time series data from: `https://www.bgc-jena.mpg.de/wetter/` and weather report from `https://www.timeanddate.com/weather/germany/jena/`

1404 `historic`. We will publish this dataset with CC BY-NC-SA 4.0 licence since the data source
1405 forbids commercial use.
1406

1407 ### L.4.2 MOTIVATION
1408

1409 The Weather-captioned dataset is designed as a semi-controlled environment to rigorously test the
1410 model's ability to learn causal relationships between text and time series, as well as its text-guided
1411 channel independence and generalizability.

1412 Such scenarios are commonly encountered in industrial applications, where correlated text and time
1413 series data often coexist. However, obtaining and releasing industrial datasets is challenging due to
1414 intellectual property restrictions. To address this, we chose the weather system—a widely available,
1415 well-understood, and publicly accessible domain—to simulate these scenarios.

1416 As an off-the-shelf TGTSF dataset, the Weather-captioned dataset provides a benchmark for evaluating
1417 the model's capacity to learn causal relationships between text and time series patterns, offering a
1418 practical and accessible alternative for research and experimentation.
1419

1420 ### L.4.3 STATISTICAL DETAIL OF THE TIME SERIES
1421

1422 For better understanding of the statistical distribution of Weather dataset, we plot the histogram of all
1423 21 channels in Fig. 11.
1424

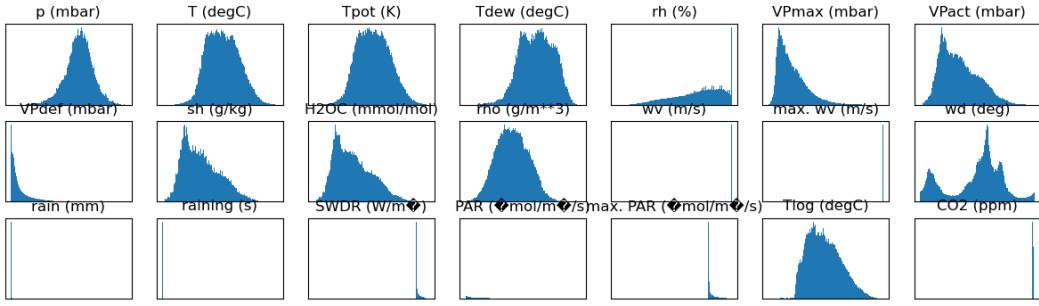

1436 Figure 11: Histogram of all 21 channels. It shows all the channels have unique value distribution,
1437 making a model hard to generalize on all channels without knowing related information.
1438

1439 ### L.4.4 USE OF LARGE LANGUAGE MODELS FOR PREPROCESSING THE WEATHER DATASET
1440

1441 We would like to clarify that the use of Large Language Models (LLMs) in this work is strictly
1442 limited to the preprocessing and creation of the Weather-Captioned dataset. LLMs are not part of our
1443 model or method, nor do they contribute to the training or inference process of TGForecaster. The
1444 Weather-Captioned dataset is intended to serve as an off-the-shelf, text-time synchronized benchmark
1445 dataset with raw text and pre-embedded text embeddings as optional inputs.

1446 The primary reason for using LLMs to preprocess this dataset is to generate diverse and correlated
1447 textual descriptions, ensuring a richer corpus for training and evaluation. By incorporating varied
1448 expressions, we enable the model to generalize to different textual forms while aligning the semantic
1449 meaning of text with time series patterns. For instance, the descriptions "The morning will be sunny,
1450 but clouds will increase in the afternoon with a chance of light rain" and "The day starts with clear
1451 skies, gradually turning cloudy with some rain in the afternoon" carry the same semantic information
1452 but differ in expression. This diversity enhances the robustness of the benchmark and validates the
1453 model's generalization capabilities.

1454 Additionally, the raw data source for this dataset often includes general weather reports in text
1455 form, accompanied by coarse numerical updates every six hours. While numerical values such as
1456 {High_Temp: 25, Low_Temp: 20, Temp_Trend: slightly increasing, Wind_Speed: 5, Wind_Direction:
1457 East} are available, they lack the precision required for reliable exogenous variables. Moreover, the
raw text contains rich semantic details—such as qualitative weather descriptions—that cannot be

effectively captured using numerical values or one-hot encoding. Using text embeddings allows the model to leverage both semantic and numerical information more effectively.

In summary, the LLM preprocessing step is solely for dataset preparation and corpus diversity, ensuring that the Weather-Captioned dataset is suitable for evaluating text-guided time series forecasting models. Our method does not rely on any LLM capabilities, and the inclusion of LLM-generated text is not a necessary step for TGTSF or any similar model. We will include raw data samples in the final paper to provide greater clarity and avoid any misunderstandings.

### L.4.5 PROMPT

To transform this quantitative data into actionable insights for TSF, we employed GPT-4 to summarize each forecasting report into a set of seven thematic sentences, as demonstrated in Tab. 1. GPT-4 is prompted to avoid specific numerical details from the original reports to prevent information leakage and to simplify the model's learning process. To further enrich the dataset, GPT-4 generated three distinct summary versions for each report, resulting in $3^7 = 2187$ possible unique textual captions for each time step. The full details of the prompt used for generating these summaries are provided in the appendix.

We generate the summary of climate report using following prompt on GPT4.

```
You are a professional weather forecast message writer. You are
↪  provided with the weather forecasting results in the next 6
↪  hours and you should transcribe it as readable text.The
↪  weather forecasting results are given in json string format.
↪  One is the coarse grained weather of the next 6 hours and the
↪  fine grained json string contains the weather forecast every
↪  half hour in these 6 hours.

You are suppose to summarise the weather forecast message in the
↪  following aspects, each aspect should be a sentence or a
↪  phrase:

1. Time of Day, Month, e.g. "It's the early morning of a day in
↪  December."
2. Current overall Weather Condition, you may use the term in
↪  coarse grained information, e.g. "The current weather is
↪  clear."
3. Weather Trend in the 6 hours, you may summarise this according
↪  to the fine grained information, e.g. "The weather is expected
↪  to remain clear." / "Rain is expected soon."
4. Temperature Trend in the 6 hours, you may summarise this
↪  according to the fine grained information, e.g. "The
↪  temperature is showing a mild drop."
5. Wind Speed and Direction, you may summarise this according to
↪  the fine grained information, e.g. "There is Light Breeze from
↪  NNW."
6. Atmospheric Pressure, describe the pressure of the atmosphere,
↪  e.g. "The atmospheric shows very Low Pressure."
7. Humidity, describe the humidity of the atmosphere, e.g. "The
↪  humidity is very high."

The summary do not have to be very detailed, but should be clear
↪  and concise.

Note that, during summarization, you should NOT include the exact
↪  values of the weather forecast, but only the trends and
↪  conditions. You should follow the following ranking
↪  instructions:
```

```
1. Time of Day: 00:00 - 06:00 -> Early Morning, 06:00 - 12:00 ->
↪ Morning, 12:00 - 18:00-> Afternoon, 18:00 - 24:00 -> Evening
2. Wind Direction: you should convert the wind direction of
↪ degrees to N, E, S, W, NE, SE,SW, NW, NNE, ENE, SSE, WSW, NNW,
↪ ESE, SSW, WNW.
3. Wind Speed: you should convert the wind speed to: Less than 20
↪ km/h -> Light Breeze, 20 to 29 km/h -> Gentle Breeze, 30 to 39
↪ km/h -> Moderate Breeze, 40 to 50 km/h -> Fresh Breeze, 51 to
↪ 62 km/h -> Strong Breeze, 63 to 74 km/h -> High Wind, 75 to 88
↪ km/h -> Gale,89 to 102 km/h -> Strong Gale, Over 102 km/h ->
↪ Storm.
4. Atmospheric Pressure: you should convert the atmospheric
↪ pressure to: (<990 mbar) -> Very Low Pressure, (990-1009 mbar)
↪ -> Low Pressure, (1010-1016 mbar) -> Average Pressure,
↪ (1017-1030 mbar) -> High Pressure, (>1030 mbar) -> Very High
↪ Pressure.
5. Humidity: you should convert the humidity to: (<30%) -> Very
↪ Dry, (30-50%) -> Dry, (51-70%) -> Average Humidity, (71-90%)
↪ -> Humid, (>90%) -> Very High Humid. You may change this to
↪ more oral expression.
6. Trend: you may use "increase", "decrease", "remain", "steady",
↪ "Go up/down"... to describe the trend of the weather
↪ condition, temperature, wind speed, atmospheric pressure, and
↪ humidity. You may change the expression.

Note that, the unit of the weather forecast may not provided, you
↪ should use the following units:
Temperature: Celsius, Wind Speed: km/h, Atmospheric
↪ Pressure/barometer: mbar, Humidity: %, wind direction: degree
↪ from 0 to 360 with 0 as North.

Following are some examples of the input and output of the task,
↪ you can make slightly changes to the output to make it more
↪ natural and fluent, but keep the main information, concise and
↪ the ranking instructions in mind:

 Example Input 1:

 xxxxxxx

 Example Output 1-1:

 It's the early morning of a day in January.
 The current weather is clear.
 The weather is expected to remain clear.
 The temperature is showing a mild drop.
 There is Light Breeze from NNW.
 The atmospheric shows Average Pressure.
 The humidity is very high.

 Example Output 1-2:

 It's the early morning of a day in January.
 The current weather is clear.
 The weather will keep clear.
 The temperature is dropping mildly.
 There is Light Breeze from NNW.
```

```
The atmospheric pressure is average.
The air is very humid.
```

### L.4.6   CHANNEL DETAILS

The meaning of each channel are as follows. The original weather dataset only contains the abbreviation for each channel, to further enrich the semantic for accurate information, we add a line of explanation after it as the channel description.

- p (mbar): Atmospheric pressure measured in millibars. It indicates the weight of the air above the point of measurement.
- T (degC): Temperature at the point of observation, measured in degrees Celsius.
- Tpot (K): Potential temperature, given in Kelvin. This is the temperature that a parcel of air would have if it were brought adiabatically to a standard reference pressure, often used to compare temperatures at different pressures in a thermodynamically consistent way.
- Tdew (degC): Dew point temperature in degrees Celsius. It's the temperature to which air must be cooled, at constant pressure and water vapor content, for saturation to occur. A lower dew point means dryer air.
- rh (%): Relative humidity, expressed as a percentage. It measures the amount of moisture in the air relative to the maximum amount of moisture the air can hold at that temperature.
- VPmax (mbar): Maximum vapor pressure, in millibars. It represents the maximum amount of moisture that the air can hold at a given temperature.
- VPact (mbar): Actual vapor pressure, in millibars. It's the current amount of water vapor present in the air.
- VPdef (mbar): Vapor pressure deficit, in millibars. The difference between the maximum vapor pressure and the actual vapor pressure; it indicates how much more moisture the air can hold before saturation.
- sh (g/kg): Specific humidity, the mass of water vapor in a given mass of air, including the water vapor. It's measured in grams of water vapor per kilogram of air.
- H2OC (mmol/mol): Water vapor concentration, expressed in millimoles of water per mole of air. It's another way to quantify the amount of moisture in the air.
- rho (g/m³): Air density, measured in grams per cubic meter. It indicates the mass of air in a given volume and varies with temperature, pressure, and moisture content.
- wv (m/s): Wind velocity, the speed of the wind measured in meters per second.
- max. wv (m/s): Maximum wind velocity observed in the given time period, measured in meters per second.
- wd (deg): Wind direction, in degrees from true north. This indicates the direction from which the wind is coming.
- rain (mm): Rainfall amount, measured in millimeters. It indicates how much rain has fallen during the observation period.
- raining (s): Duration of rainfall, measured in seconds. It specifies how long it has rained during the observation period.
- SWDR (W/m²): Shortwave Downward Radiation, the amount of solar radiation reaching the ground, measured in watts per square meter.
- PAR (umol/m$\hat{2}$/s): Photosynthetically Active Radiation, the amount of light available for photosynthesis, measured in micromoles of photons per square meter per second.
- max. PAR (umol/m$\hat{2}$/s): Maximum Photosynthetically Active Radiation observed in the given time period, indicating the peak light availability for photosynthesis.
- Tlog (degC): Likely a logged temperature measurement in degrees Celsius. It could be a specific type of temperature measurement or recording method used in the dataset.
- CO2 (ppm): Carbon dioxide concentration in the air, measured in parts per million. It's a key greenhouse gas and indicator of air quality.

### L.4.7 VISUALIZATION OF THE TEST SAMPLE

We show a segment of test sample along with the dynamic news timeline in Fig. 12. The news messages are sparse and vague and not directly correlated to some of the channels. These text are passed to the model as text embeddings and aligned with time series on time domain. Thus, the model can extract causal relationship to guide each channel to perform accurate prediction even though they have distinguished distribution.

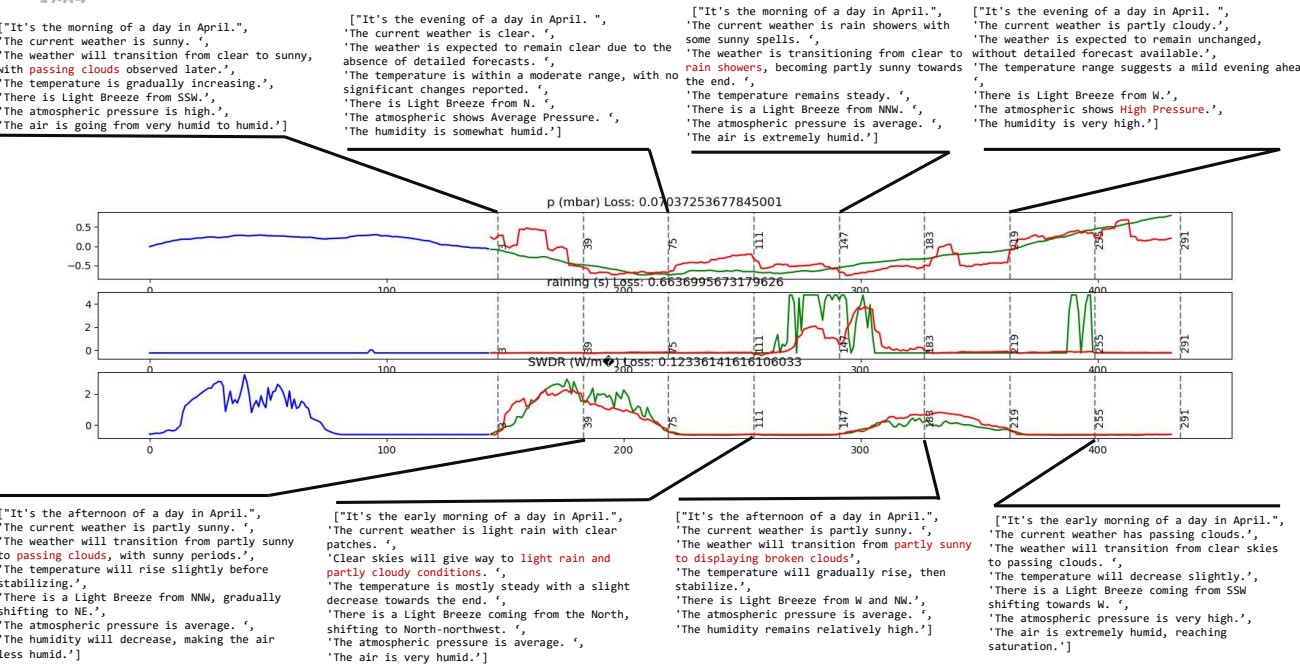

Figure 12: An visualization of test sample with all the corresponding dynamic news. Weather-Captioned dataset have dynamic weather report update every 6 hours. As we demonstrate a case of predicting 48 hours. Note that the embedding of these sentences are fed to the TGForecaster along with the look-back window time series as input. We highlight some of the words that may make impact on the forecasting result.

The following sections give detailed performance and visualization across all the channels.

### L.5 STEAM DETAILS

We are not directly publishing this dataset because of the intellectual property restrictions.

For those who are interested in reproducing the dataset: We directly crawl all the event data on Steam News of each Game. We may release the script for this later.

As for the online players, we use a randomly picked publicly available database that provide the downloading of csv file containing online gamer number. We will not release any data or tools to get these data.

## M IMPLEMENTATION DETAILS AND HYPER-PARAMETERS

We train our model on single NVIDIA A800 GPU.

For electricity dataset, we directly report the result from the original paper. For weather dataset, we uses the exact set of hyper-parameter for the original weather datasets provided by each baseline model.

In most of the experiments, we simply use a patch length of 6 and stride of 3. For Toy dataset, we use patch length of 16 and stride of 8.

We follow the previous works, split all the dataset by 7:1:2 for training, validation and testing.

Except the performance on the Weather-Captioned Dataset, all other experiments are ran on the MiniLM Embedding. We selected MiniLM as the embedding model because it achieves results comparable to OpenAI embeddings while producing smaller embeddings (384 dimensions for MiniLM versus 512 for OpenAI). This reduced embedding size speeds up training, particularly for ablation studies, making it more practical for our experiments.

Further detailed hyperparameter settings are provided in the training scripts in our codebase. We did not perform comprehensive hyper-parameter tuning because of the constraint of compute power. Thus, we may report a sub-optimal result of TGForecaster.

