# OpenReview forum: "Beyond Trend and Periodicity: Guide Time Series Forecasting with Textual Cues"
_ICLR.cc/2025/Conference — ICLR 2025 Conference Withdrawn Submission_

### Official Review · Reviewer_GhyY · 2024-10-16

**Soundness:** 3
**Presentation:** 3
**Contribution:** 2
**Rating:** 6
**Confidence:** 4

**Summary:**

This paper introduces a novel Text-Guided Time Series Forecasting (TGTSF) task, integrating textual cues, such as channel descriptions and dynamic news, to enhance traditional forecasting methods that rely solely on historical data. The authors propose TGForecaster, a robust baseline model employing cross-attention mechanisms to fuse textual information with time series data. They present four meticulously curated benchmark datasets, ranging from simple periodic patterns to complex, event-driven fluctuations, to validate this task. Comprehensive evaluations demonstrate that TGForecaster consistently outperforms existing methods, highlighting the significant potential of incorporating textual information into time series forecasting.

**Strengths:**

1. This paper introduces a novel task for multimodal time series analysis.
2. This paper presents four comprehensive multimodal time series datasets.
3. The extensive experiments look hopeful.

**Weaknesses:**

While this paper explores the potential of multimodal time series analysis, I have the following concerns:

1. **Information leakage**: Although the paper repeatedly emphasizes that their text data is known prior to prediction and that no information leakage occurs, I remain unconvinced. Among the three text forms, I believe only common knowledge is likely secure. System-level limited predictions and hypothesized or controlled events both present potential risks of information leakage.

   - Equation 1 indicates that TGForecaster relies on textual information that not only pertains to the future but is also precisely aligned with the point in time being predicted.
   - For system-level limited predictions, as exemplified by Figure 6, we must know not only that it will rain, but also the exact time the rainfall will cease in order to accurately capture rainfall peaks.
   - For hypothesized or controlled events, such as the rainfall prediction in row 220, we must not only anticipate artificial rainfall but also the duration of its impact.

   It seems challenging to provide this level of information using non-leaking text, which significantly limits the practical applicability and scope of the paper.

2. **Dataset contributions**: The authors claim to have contributed four datasets. Setting aside the synthesized dataset, we are left with three:

   - **Electricity**: Only holiday information is introduced as external text, and there seems to be no clear advantage over previous approaches utilizing timestamps [1].
   - **Weather**: From Appendix N.3, it appears they employed GPT-4 to convert accurate numerical predictions into textual predictions. Since precise numerical forecasts are already available, a direct approach using exogenous variables seems more efficient.
   - **Steam**: This dataset is not publicly available due to intellectual property restrictions.

   In summary, these datasets fail to convincingly demonstrate the utility of the paper.

3. **Task novelty**: The authors assert that they have introduced a text-guided time series forecasting task, yet previous works like Time-MMD [3] exist. While I do not expect the authors to conduct a quantitative comparison, given that both works may belong to the same period, I encourage them to discuss the distinctions between the two as they deem appropriate. To my knowledge, Time-MMD only utilizes historical textual information and does not face the issue of information leakage. Furthermore, its dataset construction appears to be more rigorous.

**References**

[1] 2024, Rethinking the Power of Timestamps for Robust Time Series Forecasting: A Global-Local Fusion Perspective

[2] 2024, TimeXer: Empowering Transformers for Time Series Forecasting with Exogenous Variables

[3] 2024, Time-MMD: A New Multi-Domain Multimodal Dataset for Time Series Analysis

**Questions:**

The authors can refer to the weakness listed above.

---

> ### Author Response · Authors · 2024-11-19
>
> Thanks for your detailed and insightful review. We hope the following response can address your concerns.
>
> - Information Leakage and Temporal alignment
>
> Within the framework of traditional time series forecasting, using such information might indeed be considered information leakage, as it incorporates causal insights. However, TGTSF operates under a different assumption: that this information is accessible at the start of forecasting. This scenario is realistic and more practical for many real-world applications, such as sales prediction, where accurate forecasts often require additional inputs beyond historical data. As discussed in the novelty section of our common response, this makes TGTSF more applicable than purely time-series-based forecasting methods.
>
> The use of exogenous variables known in advance to aid forecasting is also not a new concept. For example, the M5 dataset includes features such as pricing, promotions, and special events that are available beforehand. TGTSF builds on this idea by introducing text as a semantically richer form of exogenous input, enabling deeper contextual understanding.
>
> Technically, TGTSF does not leak information directly derived from the time series. As explained in the common response, the textual inputs are gathered from external sources and do not involve generating captions from the time series data. Furthermore, the text does not describe specific time series directly but instead outlines the overall state of the system. TGTSF leverages TGCI to extract causal relationships from the text and map them to specific channels, ensuring a clean separation between textual inputs and time series data.
>
> For the precise alignment, in the dataset we used the text information comes in very coarse granularity, the electricity and steam dataset comes with granularity of day and weather-cap have update every 6 hours. This is very coarse comparing to the time series sampling rate we are forecasting. Thus, the precise alignment is not necessary. This also highlight our use case of upsampling the coarse prediction both on time domain and channel with limited news text input.
>
> That said, for scenarios requiring highly precise temporal alignment, such as high-frequency stock trading with sub-second sampling intervals, this approach may not be ideal. In such cases, numerical exogenous variables or other methods might be more appropriate. TGTSF is more suitable for applications where coarse-grained inputs, like general system states or external news, provide actionable insights without requiring exact temporal precision.
>
> - Dataset Design
>
> For the Electricity dataset, the text input includes straightforward information such as the day of the week and whether it is a public holiday. This is equivalent to what can be derived from timestamps, so it is expected that the results would be similar to timestamp-based methods.
>
> For the weather-captioned dataset, we specifically designed it to test TGTSF models' ability to learn causal relationships. As shown in the “Use of Text” section of the common response, our model need text and time series that have a strong causal relationship to learn the correct causal relationship. Weather forecasts, which are generally accurate and widely available, provide an ideal source for generating such causal and correlated text. We used GPT to create captions primarily to introduce diversity in the text, ensuring a richer corpus for the model to learn from. While numerical weather data could be used as exogenous variables, some information—such as general weather conditions or special events—cannot easily be expressed numerically. Text provides a semantically rich input that allows the model to extract correlations that are not directly encoded in numbers, such as associating sunny weather with higher solar radiation levels.
>
> We acknowledge that the weather-captioned dataset is a semi-controlled experimental setting rather than a fully realistic scenario. However, it effectively demonstrates TGForecaster’s ability to extract causal relationships and its potential to perform well when provided with accurate, correlated textual input.
>
> - Novelty comparing to the Time-MMD
>
> Yes, as mentioned by reviewer, our work and Time-MMD belongs to the same period. But Time-MMD more focus on the traditional time series definition where they only use previous text to enhance the forecasting. On the other hand, our TGTSF propose to extract the causal relation between event in text and time series patterns. Although we all focus on integrating text information in time series forecasting, we are taking a slightly different route and interpretation of the task. We will add this analysis to the related work section in the final version of the paper. For more detailed discussion on Novelty, please check the Novelty section of the common response.

---

> > ### Comment · Reviewer_GhyY · 2024-11-20
> >
> > 1. **Information Leakage**
> >
> >    The use of coarse-grained textual information does indeed mitigate the risk of information leakage, and I largely agree with the authors' explanation. However, I have another concern. How does the model achieve horizontal and vertical alignment between coarse-grained text and numerical values? Taking the fourth textual prompt in Figure 6, *"Clear skies will give way to light rain and partly cloudy conditions,"* as an example, how does the model determine which hour or even minute within the next six hours the rainfall starts to increase (horizontal alignment), and by how much (vertical alignment)?
> >
> > 2. **Dataset Contribution**
> >
> >    There seems to be no controversy regarding the insufficient contributions of the Toy, Electricity, and Steam datasets. For the Weather dataset, the limited discrete sets representing sunny weather or other general weather conditions could be encoded using one-hot embedding. Regarding special events, these indeed serve as a strong motivation for introducing textual information. However, as shown in Table 1 and Appendix N.3, the dataset in this paper do not appear to include such events. According to N.3, GPT-4 was used to transform precise quantitative data into textual information. I am perplexed about the advantage of this step. This transformation seems to result in information loss, and directly using exogenous variables might be more effective.
> >
> > Furthermore, it is recommended that the author improve the visual quality of the rough charts in the article. Figure 6 lacks a legend for the predicted and actual values. In Figures 6, 8, and 12, the text overlaps with the bounding boxes, making them difficult to distinguish. Moreover, there are even illegal characters present in Figure 12.

---

> > ### Author Response · Authors · 2024-11-27
> >
> > Dear Reviewer,
> >
> > We would like to sincerely thank you for your thoughtful feedback and suggestions on our manuscript. In response, we have updated the manuscript, incorporating several of your suggestions and addressing the concerns you raised. Specifically, we have added a section to clarify that our dataset is designed as an off-the-shelf text-time series dataset, requiring no further LLM captions, which we hope will help alleviate any potential confusion. In the same section, we also explain the motivation for using the weather-captioned dataset as a proxy benchmark to test TGTSF’s capabilities. Additionally, we have included an updated ablation study that further clarifies the key mechanisms behind our design, and we have added a new background section discussing text embedding techniques and related works.
> >
> > Also, we have rescaled and reordered some figures to make them more readable. We are considering replace some of them with regenerated version in the final version.
> >
> > We believe these revisions provide a clearer understanding of our approach and its novelty. If these clarifications address your concerns, we would respectfully request that you reconsider the score.
> >
> > Thank you for your time and consideration.
> >
> > Sincerely,
> > The Authors

---

> ### Author Response · Authors · 2024-11-20
>
> Thank you for your quick response.
>
> -  Precise Horizontal and Vertical Alignment
>
>     We do not guarantee precise horizontal or vertical alignment in the input data. Providing such detailed information—for example, exactly when it will rain and how much—would constitute severe information leakage. Instead, the model forecasts time series patterns based on its training. In the example you mentioned, the model might have encountered similar text during training and learned the corresponding pattern, such as a mild temperature drop when the weather changes from sunny to light rain, accompanied by increased humidity. This is what the model is designed to learn and predict: generating precise numerical time series forecasts guided by coarse but causal textual inputs.
>
>     Furthermore, the model does not rely solely on a single piece of dynamic news. It also incorporates other contextual information, such as the season and time of day. By combining all these inputs, the model determines the time series patterns it forecasts.
>
> - Dataset Contribution:
>
>     We acknowledge that the Toy and Electricity datasets may appear simple and straightforward. However, as mentioned in the Dataset Section of our manuscript, when considered together with the other datasets as part of the TGTSF benchmark, they represent a progression of increasing difficulty and complexity. Each dataset is carefully designed to evaluate the model’s capabilities from different perspectives. First, the toy dataset is designed with most straightforward and simple pattern, but without text, the model cannot predict the sudden change of the pattern. This test the model's ability to adhere the guidance of text. The electricity dataset is designed to test the model's ability to predict the pattern with the guidance of simple text. Then because this is a commonly-used off-the-shelf dataset with common knowledge as text, we can easily compare with other existing method on it. **Of course, we need large, comprehensive and complex dataset to train a model to show its superior performance. We also need simple and easy-to-control dataset to help us validate and understand the model's behavior.**
>
>     The Weather-captioned dataset serves as a semi-controlled environment to rigorously test the model’s ability to learn causal relationships between text and time series, as well as its text-guided channel independence and generalizability. Lastly, the Steam dataset provides a fully real-world scenario, showcasing TGTSF’s superior performance in practical applications. Although we cannot release this dataset due to intellectual property restrictions, we will provide detailed instructions in our code repository for replicating it.
>
>     As the first benchmark for TGTSF, we want to provide a comprehensive evaluation method from multiple levels, i.e. from simple to complex, from controlled and comprehensible to real-world and complex. This benchmark is not only designed as a ranking tool but also as a means to help researchers analyze and improve their models’ behaviors. We see this as a starting point for the community and hope that more researchers will contribute datasets to expand and enhance this benchmark in the future.
>
> - Use of GPTs on weather dataset:
>
>     GPT is used only to generate diverse and correlated text for the weather dataset, so that we can treat it as a time series dataset that really comes with highly correlated textual weather report. However, **GPT usage is not part of our proposed method and does not contribute to TGTSF’s performance**. Therefore, we do not highlight it as an advantage of our approach.
>
>     In the context of our paper, the weather-captioned dataset is treated as an off-the-shelf TGTSF dataset that inherently provides aligned text and time series data. The detailed information we provide about creating this dataset is purely for transparency and reproducibility, similar to what we provide for other datasets.
>
> - One-hot Encoding:
>
>     We acknowledge that one-hot encoding can be used to represent general weather conditions. However, as noted in the first question, real-world weather data often includes complex and diverse textual descriptions, such as *“Clear skies will give way to light rain and partly cloudy conditions.”* Representing such nuanced text with one-hot encoding would require either an impractically large number of categories or significant simplification, which would result in the loss of valuable information.
>
>     Furthermore, the weather-captioned dataset is designed as a dataset that directly provides text alongside time series data. Using one-hot encoding in this context would be labor-intensive and imprecise. Instead, we use text embeddings to represent the text and allow the model to learn the correlations between textual semantics and time series patterns in a fully automated and scalable manner. This approach also ensures better adaptability to more diverse and complex real-world scenarios.

---

> ### Author Response · Authors · 2024-11-20
>
> - About chart quality:
>
>     We do apologize for the rough charts in the Appendix. We will improve the visual quality of the figures in the final version of the paper. We will ensure that the figures are clear, legible, and free of any errors. For the illegal characters in Figure 12, it comes with the dataset itself, and it seems to be an encoding problem. To maintain the originality, we decide to keep it.
>
> Looking forward to further discussion with you!

---

> ### Comment · Reviewer_GhyY · 2024-11-21
>
> What concerns me is that, whether it’s the Electricity dataset or the Weather dataset, the so-called “complex and diverse textual descriptions” presented by the authors are derived from precise numerical data that could have been directly utilized through exogenous variable methods. If the text information originated from native sources like breaking news events (like Time-MMD), the value of the dataset would increase significantly. However, this current transformation from precise numerical data to vague textual descriptions inevitably leads to information loss, giving me the impression of unnecessary complication. Admittedly, one-hot embedding cannot accurately represent cases like “*Clear skies will give way to light rain and partly cloudy conditions*”, but this text itself is a summary derived from the precise numerical data of weather forecasts. So, why not directly use the original precise numerical data instead?

---

> ### Author Response · Authors · 2024-11-21
>
> Thanks for your quick response and raising these insightful questions.
>
> - What is the weather-captioned dataset for?
>
>     *The purpose of the weather-captioned dataset is to simulate a scenario for testing the TGTSF model’s capabilities rather than solving the weather forecasting problem itself.* We acknowledge that our weather data source includes certain numerical values, which could theoretically be used as exogenous variables (as discussed later). However, the motivation behind this dataset is to create a controlled environment featuring time series data sampled from a physical system alongside correlated text that describes the system’s dynamics or events.
>
>     This type of scenario is common in industrial applications, where such data often exists. However, industrial datasets are challenging to obtain and release due to intellectual property restrictions. Therefore, we chose the weather system—a well-known, widely available, and understood system—to simulate this scenario. As an off-the-shelf TGTSF dataset, the weather-captioned dataset serves as a benchmark to evaluate the model’s ability to learn causal relationships between text and time series data.
>
>     To illustrate, this approach is akin to building a rocket for a real-world problem: while constructing a rocket for every test is infeasible, engineers use CAD simulation tools to validate designs and ideas. Similarly, the weather-captioned dataset simulates the text-guided forecasting problem, allowing us to validate our approach.
>
>     Moreover, benchmarks are not designed to solve specific real-world problems directly but to serve as proxy metrics for evaluating models on a set of tasks. In this sense, the TGTSF benchmark remains significant and valuable.
>
> - Why convert the "numerical values" to text in our case?
>
>     Beyond our design considerations, the raw data source for this dataset does not always provide **precise numerical values**. Typically, the data consists of general daily weather reports in text form, such as:
>     *“The morning will be sunny, but clouds will increase in the afternoon with a chance of light rain. Temperatures will be around 20-25 degrees, slightly lower than yesterday. Light breeze from the east at 5-10 km/h.”*
>
>     This text may be accompanied by numerical updates every six hours, such as:
>     *{High_Temp: 25, Low_Temp: 20, Temp_Trend: slightly increasing, Wind_Speed: 5, Wind_Direction: East}*
>
>     While these numerical values exist, they are not precise enough to serve as reliable exogenous variables. Additionally, the text includes rich semantics—such as qualitative weather descriptions—that cannot be easily represented by numerical values or one-hot encoding. Therefore, we use text embeddings to represent the text, allowing the model to leverage both the numerical and semantic information effectively as exogenous input. We will include a raw data sample in the final version of our paper to avoid such confusion.
>
> - In the scope of weather forecasting, if it is better to use percise numerical values as exogenous variables?
>
>     Setting aside the purpose of the weather-captioned dataset, we can consider the general feasibility of using numerical values as exogenous variables for weather forecasting.
>
>     In the context of weather forecasting, precise numerical values from weather data can indeed serve as effective exogenous variables and may lead to performance improvements. For researchers with access to detailed meteorological data, building models with such precise inputs (e.g., ClimODE [1]) is certainly feasible. However, for those who only have access to general weather reports, the numerical values provided are often vague or inaccurate.
>
>     Numerical inputs are inherently strong and deterministic, which makes models highly sensitive to the quality of the original forecast. In contrast, text, while less precise, provides semantic richness and is relatively robust to inaccuracies in the forecasts. By using vague text as guidance, the model can reason through the semantics to infer precise values based on the historical time series data.
>
>     This trade-off between precision and robustness is an interesting question, and we plan to investigate it further in future work.
>
>     [1] ClimODE: Climate and Weather Forecasting with Physics-informed Neural ODEs
>
> I hope these can address your concerns and looking for further discussion.

---

> ### Comment · Reviewer_GhyY · 2024-11-21
>
> The author has basically resolved my concerns, and I will improve my score.

---

> > ### Author Response · Authors · 2024-11-21
> >
> > Thank you for recognizing our work! This has been a truly insightful discussion, highlighting some overlooked aspects in the design of our benchmark. We greatly appreciate your feedback and will incorporate these insights into the updated version of the paper.
> >
> > We are happy to continue the discussion if you have any further questions or suggestions!

---

### Official Review · Reviewer_Xiwt · 2024-10-16

**Soundness:** 2
**Presentation:** 2
**Contribution:** 2
**Rating:** 3
**Confidence:** 5

**Summary:**

The paper addresses a common challenge in time series forecasting (TSF): information insufficiency. It proposes an approach, TGForecaster with open-source word embedding, transformer block and cross-attention mechanism, which integrates textual cues to enhance forecasting models with external knowledge. The authors developed four datasets designed to validate TGTSF task. The proposed TGForecaster model demonstrates improvements in toy dataset and perform as good as previous small-size work in other dataset.

**Strengths:**

Thanks for your work! The following are some strengths:
1. The paper is well organized and the motivation is clear
2. The paper shows a good result especially on toy dataset, which proves that it at least work in some way.
3. The author give ablation study in their paper which compare open-source embedding and w/o some texts, which is direct and clear

**Weaknesses:**

1. TGForecaster itself is not innovative, which use open-source embedding and attention to do information fusion.  As far as I know, numbers derive from sampling, words come from natural description, so it's important to decide how to fuse the information with a more well-designed transformer or better alignment strategy, otherwise I would say this good result is based on the raise of parameters number. Also, for channeling modelling, graph-related work also give some insights.
2. The ablation study is just too simple, which leaves people thinking about \textit{what is the role of these words in TGForecaster ?}. This question is not about with or without words, but what will happen if I give random words. Your work involve long sequence of texts, which give many high-dimension variables and regression from there variables and give some ordinary result is simple.
3. It's really new for multi-modal time-series tasks, but the texts are not well-designed and are simply collect together, so I suggest doing more jobs on  a. how to fairly compare  b. how to align  c. how to evaluate the importance of texts and select  d. dig some real theory behind, eg. how to extract the essential information in high-dimension or how to compress unfixed texts
4. Limited evaluation dataset.
5. weak generalization ability.

**Questions:**

1. How did you test Time-LLM(Jin et al., 2023), did you train it with your designed texts, can you give some samples how you evaluate the performance of Time-LLM and which language model did you use.
2. I wonder if there is some information leakage. For example, in the prediction of weather, words may indicate the time-series trend or related property
3. Are TGForecaster really able to understand the text? How do you study this.

---

> ### Author Response · Authors · 2024-11-20
> **Response #1**
>
> We thank reviewer's insightful questions and suggestions. Hope the following response can address your concerns.
>
> - Performance not just as good but significantly better than previous methods:
>
>     The Toy dataset was specifically designed to be unpredictable without textual cues, so the inclusion of correlated text naturally leads to a significant performance boost. This aligns with the dataset’s purpose.
>
>     Furthermore, TGForecaster demonstrates remarkable performance improvements on other datasets, particularly the Weather dataset. For instance, while PatchTST improved performance by 8.3% and TimeLLM by 1.6% compared to the previous state-of-the-art, TGForecaster achieves an impressive **32%** improvement. Additionally, our method delivers over **80%** accuracy improvement on specific unpredictable channels, such as atmospheric pressure (as shown in Table 6, Appendix E). On the highly noisy and random Steam dataset, TGForecaster achieves over a 12% performance boost.
>
>     **Our work demonstrates a significant breakthrough not only in achieving substantial absolute performance improvements but also in the magnitude of improvement over the state-of-the-art, highlighting the novelty and effectiveness of TGTSF and TGForecaster.**
>
> - TGForecaster itself is innovative:
>
>     As the first model designed for text-guided time series forecasting, TGForecaster introduces a novel paradigm. Similar to how BERT [4] established masked language modeling and validated its feasibility with a simple transformer, TGForecaster paves the way for broader adoption of this task through its focused design.
>
>     **Innovation in TGForecaster lies not in creating new computational blocks but in their insightful application.** Specifically, the model uses cross-attention in two unique mechanisms tailored to this task, enabling it to capture causal relationships between text and time series patterns effectively. These mechanisms are discussed in the manuscript, and we will further elaborate on their motivations in the final paper. For a detailed explanation of the model and task novelty, please refer to the “Novelty” section in the Common Response.
>
> - Performance boost does NOT because of more parameters
>
>     Our ablation studies clearly show that without textual guidance, TGForecaster’s performance reverts to a level comparable to PatchTST. This demonstrates that the performance improvement comes from the inclusion of text guidance rather than additional parameters. The results are detailed in the ablation section of the manuscript.
>
> - Diverse text enables generalization:
>
>     Our approach does not rely on well-designed text; instead, the model aligns with the “text embedding space” or “text semantic space.” In this space, sentences with the same meaning are placed close together, ensuring TGTSF’s generalizability to raw, diverse text inputs. This property is inherent to the pre-trained text embeddings we use. As a result, text can be directly transformed into embeddings and fed into TGForecaster, enabling a fully end-to-end text-guided time series forecasting solution.
>
>     In fact, one reason we use GPT to generate text is to introduce diversity. The goal is for the model to learn that different sentences with similar semantics can correspond to the same time series pattern. This adaptability to diverse textual inputs is evident in the results on the weather-captioned dataset, where test samples use randomly selected texts with varying expressions but identical meanings. The observed error bars reflect the model’s ability to generalize effectively across these variations.
>
> - Evaluation of Time-LLM
>
>     For a fair comparison, we use the official release of the Time-LLM. We maintain consistency by using the same channel descriptions for Time-LLM. However, in testing the baseline Time-LLM, we adhered to their established pipeline for generating prompts, which does not incorporate our dynamic news. This approach ensures that our comparative analysis remains fair and within the original operational framework of the baseline model.
>
> - Modelling the channels is already studied with graph-based methods
>
>     Unlike graph-based methods that model inter-correlations among channels, TGCI focuses on capturing the correlation between channels and dynamic news. TGCI is specifically designed with two key objectives:
>
> 	1.	To create universal and semantically meaningful identifiers for channels that reflect their unique distributions.
>
> 	2.	To capture correlations between text and channels while considering the distinct responses of different channels to the same input.
>
>     TGCI achieves these objectives by leveraging semantic channel descriptors to guide the extraction of relevant information from textual inputs, offering a novel approach distinct from graph-based techniques.

---

> ### Author Response · Authors · 2024-11-20
> **Response #2**
>
> - Ablation about random words:
>
>     This is a really good ablation that we overlooked in the manuscript and it can evidently reveal the mechanism how TGForecaster make use of these text embeddings. We have conduct this experiment and we report the findings below.
>
>     TGTSF is designed to learn causal relationships between events described in text and their corresponding time series patterns. While not explicitly an alignment model, it effectively aligns the semantic meaning of text with the time series data it impacts. The model generates time series patterns guided by the textual information, and its performance varies based on the quality of the text input:
>
>     1. Training with Meaningful and Relevant Text:
>     - Inference with Similar Text: Produces strong results by accurately extracting causal relationships between events in the text and time series patterns.
>     2. Training with Zero/Random Text:
>     - Inference with Any Text: Produces results equivalent to PatchTST, as no additional information is present in the text. The model relies solely on the time series data, ignoring the random text.
>     3. Training with Meaningful Text, Inference with Incorrect Text:
>     - Inference with Incorrect Text: Results are poor, as the model relies on the misleading text input and generates patterns based on incorrect or irrelevant information.
>
>     We detail the TGForecaster performance under different text conditions in the following table:
>
>     |Test w.\Train w.|Good Text Emb.|Zero Text Emb.|Random Text Emb.|
>     |:--:|:--:|:--:|:--:|
>     |Good Text Emb.|Good results (captures causal relationships)|Poor results (random patterns)|Acceptable results (similar to PatchTST)|
>     |Zero Text Emb.|Poor results (repetitive patterns)|Acceptable results (similar to PatchTST)|Acceptable results (similar to PatchTST)|
>     |Random Text Emb.|Poor results (random patterns)|Poor results (random patterns)|Acceptable results (similar to PatchTST)|
>
>     We will add further statistical analysis and visualization to support these claims in the final version of the paper.
>
>     These outcomes demonstrate that TGTSF effectively achieves alignment in the “event” space, linking events described in the text to the corresponding time series patterns.
>
>     We acknowledge some overlooked points in this section and plan to address them in the final version of the paper, e.g. add text embedding technique to the related work; add the result of random embedding as our main ablation study experiment.
>
> - Concerns about Information leakage
>
>     Within the framework of traditional time series forecasting, using such information might indeed be considered information leakage, as it incorporates causal insights. However, TGTSF operates under a different assumption: that this information is accessible at the start of forecasting. This scenario is realistic and more practical for many real-world applications, such as sales prediction, where accurate forecasts often require additional inputs beyond historical data. As discussed in the novelty section of our common response, this makes TGTSF more applicable than purely time-series-based forecasting methods.
>
>     The use of exogenous variables known in advance to aid forecasting is also not a new concept. For example, the M5 dataset includes features such as pricing, promotions, and special events that are available beforehand. TGTSF builds on this idea by introducing text as a semantically richer form of exogenous input, enabling deeper contextual understanding.
>
>     Technically, TGTSF does not leak information directly derived from the time series. As explained in the common response, the textual inputs are gathered from external sources and do not involve generating captions from the time series data. Furthermore, the text does not describe specific time series directly but instead outlines the overall state of the system. TGTSF leverages TGCI to extract causal relationships from the text and map them to specific channels, ensuring a clean separation between textual inputs and time series data. Further discussion seen in the "Debatable Information Leakage" section of common response.
>
> - Does it really understand text?
>
>     As discussed in the “Use of Text” section of the common response, no, TGForecaster does not “understand” the exact text. Instead, it operates in the semantic space, aligning text semantics with time series patterns. This mechanism allows TGForecaster to focus on the meaning conveyed by the text rather than its specific form, enabling the potential generalizability described in the response to the question on "Diverse text enables generalization.”
>
> Hope these can address your concerns, and we are looking forward to further discussion.

---

> > ### Author Response · Authors · 2024-11-23
> >
> > Dear reviewer,
> >
> > We are eagerly looking forward to your response and following discussion.
> >
> > We address the key points as follows:
> > 1. Performance Beyond Previous Methods: TGForecaster achieves significant improvements, including a 32% boost on the Weather dataset, over 80% accuracy improvement on specific channels, and a 12% gain on the noisy Steam dataset. These results demonstrate breakthroughs in both absolute performance and improvement magnitude over the state-of-the-art.
> > 2. Innovation in TGForecaster: As the first model for text-guided time series forecasting, TGForecaster introduces a novel paradigm, leveraging cross-attention to align text semantics with time series patterns. Its innovation lies in how text embeddings are applied to validate the TGTSF task.
> > 3. Performance Improvements Not Due to Parameters: Ablation studies confirm that TGForecaster’s gains come from text guidance, not additional parameters. Without text, its performance reverts to levels comparable to PatchTST.
> > 4. Generalization Through Diverse Text: TGForecaster aligns semantically similar texts in the embedding space, enabling generalization to diverse textual inputs. Results on the weather-captioned dataset confirm its adaptability to varying textual expressions.
> > 5. Evaluation of TimeLLM: Using the official TimeLLM implementation, we observed poor performance on both simple (Toy) and complex (Weather) datasets, suggesting limitations in its ability to capture dynamic patterns compared to TGForecaster.
> > 6. Comparison with Graph-Based Methods: TGCI focuses on modeling correlations between text and channels, creating semantically meaningful descriptors and capturing channel-specific responses, unlike graph-based inter-channel correlation methods.
> > 7. Ablation Study with Random Words: Additional experiments show TGForecaster relies on meaningful text to capture causal relationships, with random text yielding results comparable to baselines. Detailed analysis will be included in the final paper.
> > 8. Addressing Information Leakage: TGTSF assumes text is available at the start of forecasting, avoiding information leakage while making the approach practical for real-world applications. Text is treated as an external exogenous variable, aligned with standard practices like the M5 dataset. We have had a thorough discussion about the weather-captioned dataset with reviewer GhyY, please check the detail if you are interested and it may address some of your remaining concerns.
> > 9. Understanding Text: TGForecaster aligns text semantics with time series patterns but does not “understand” text. This alignment enables robust generalization, as demonstrated by its performance on diverse textual inputs.
> >
> > We hope these responses clarify your concerns and welcome further discussion.

---

### Official Review · Reviewer_Gvc9 · 2024-11-03

**Soundness:** 3
**Presentation:** 3
**Contribution:** 3
**Rating:** 6
**Confidence:** 4

**Summary:**

This article addresses the limitations of insufficient information in time series forecasting. It introduces TGForecaster, a Transformer-based multimodal model designed to integrate channel descriptions and dynamic news texts for time series forecasting, which includes two key innovations: Time-Synchronized Text Embedding and Text-Guided Channel Independent (TGCI). To validate the effectiveness of Text-Guided Time Series Forecasting (TGTSF) task, this article proposes four types of multimodal datasets to test the model's capabilities. Experimental results show that TGForecaster consistently outperforms all SOTA models on four datasets and underscore the significance of textual data on performance enhancement.

**Strengths:**

1. The article has a clear structure, relatively sufficient experiments, and clear presentation of experimental results, and the writing is relatively smooth.
2. The TGForecaster framework proposed in this article is novel and its effectiveness has been verified through experiments.

**Weaknesses:**

1. The article mentions two key innovations of Time-Synchronized Text Embedding and Text-Guided Channel Independent, but the experimental part lacks ablation experiments on these two innovative points. Suggest the author to supplement experiments to better highlight these two innovative points.

2. For the comparative experiment part, this article lacks a comparison with the naïve model. It is recommended that you can refer to the MM-TFSlib framework mentioned in Time-MMD (https://arxiv.org/pdf/2406.08627v2), which uses existing time series forecasting models and LLMs to process time series and textual data separately, and then combines them for forecasting. Suggest the author to supplement this experiment to better illustrate the necessity and rationality of TGForecaster in the aspect of architecture design.

3. In Table 2 and Table 7 (in Appendix I), the experimental results of the electricity dataset (LBW=120 and Pred. Len.=720) are inconsistent (0.200/0.209). Please confirm and make the necessary modifications.

4. In the “Ablation Study” part on page 9, the author evaluates the impact of three different embedding models on the performance of TGForecaster based on the Weather-Captured-Medium dataset. According to the comparison with the experimental results in Table 3, the author ultimately chooses the OpenAI embedding. So why do we choose MiniLM as the embedding model when conducting comparative experiments on whether adding channel descriptions and news texts or not? Suggest the author to supplement corresponding experiments or provide explanations.

5. Regarding the confusion of the subplots "wv (m/s)" and "max. wv (m/s)" in Figure 7 (in Appendix F): according to the time series historical ground truth, theoretically they may be two easy forecasting cases and the forecasts given by PatchTST (Green line) should be close to a horizontal line, but the actual forecasts differ significantly from it. please explain the reason.

6. The color bar is missing from Figures 9 and 10 in the Appendix H. Please add it. In addition, there are duplicate words in the caption of the two figures, such as "on on" and "onon" in lines 1078 and 1114. Suggest the author carefully checks the wording in this article.

**Questions:**

Refer to "Strengths And Weaknesses".

---

> ### Author Response · Authors · 2024-11-19
>
> Thanks for your inspiring and insightful questions. Hope the following response can address your concerns.
>
> 1. Ablation study on mechanisms:
>
>    We conducted ablation studies by removing key inputs, such as dynamic news and channel descriptions, to evaluate the effectiveness of these mechanisms. Replacing channel descriptions with zero embeddings essentially disables TGCI, preventing the model from extracting correlations between the news set and specific channels. Similarly, replacing dynamic news with zero embeddings prevents the model from capturing the causal relationship between the news and time series patterns.
>
>    Additionally, we expanded the ablation study on the impact of dynamic news, as detailed in the “Use of Text” section of the common response. These results confirm that our model effectively learns the causal relationship between news and time series patterns, aligning with our expectations.
>
> 2. Comparision with MM-TFSlib.
>
>    Thank you for pointing this out. MM-TFSlib and TGTSF are contemporaneous works, developed during the same period, but are designed for slightly different tasks, which may require some adjustments to enable direct comparison. We will try exploring integrating these streamlined methods and include the results in the final version of our paper.
>
> 3. PatchTST's bizarre forecasting behavior.
>
>     We also observed this unusual phenomenon during our analysis, which aligns with one of the challenges our work seeks to address. While PatchTST performs well in capturing periodic patterns (e.g., in SWDR channels), it struggles with channels exhibiting random patterns, such as wind velocity (wv) and pressure (p), particularly in the absence of additional information.
>
>     We attribute this limitation to the “information insufficiency” inherent in purely time-series datasets. During training, PatchTST encounters diverse patterns but lacks any indicators to explain what caused them. Consequently, during inference, it continues to produce outputs with high randomness.
>
> 4. Why use MiniLM for ablation study?
>
>     We selected MiniLM as the embedding model because it achieves results comparable to OpenAI embeddings while producing smaller embeddings (256 dimensions for MiniLM versus 512 for OpenAI). This reduced embedding size speeds up training, particularly for ablation studies, making it more practical for our experiments.
>
> 5. Typos in the Appendix:
>    We apologize for the typos in the appendix and will ensure they are corrected in the final version:
>    - The 0.209 in Appendix I is a typo and should be 0.200 as the Table 2.
>    - We will fix the caption on Figure 9 and 10 and update the color bar.

---

> > ### Comment · Reviewer_Gvc9 · 2024-11-22
> >
> > Thanks for the detailed response, which adequately addresses my concerns, and I would like to improve the Confidence.

---

> > > ### Author Response · Authors · 2024-11-22
> > >
> > > Thanks for your recognition of our work. We are happy to continue the discussion if you have any further questions or suggestions!

---

> > ### Author Response · Authors · 2024-11-27
> >
> > Dear Reviewer,
> >
> > We would like to sincerely thank you for your thoughtful feedback and suggestions on our manuscript. In response, we have updated the manuscript, incorporating several of your suggestions and addressing the concerns you raised. Additionally, we have included an updated ablation study that provides further insights into the key mechanisms behind our design.
> >
> > Regarding the color bar in the attention maps, we decided not to include it. As the absolute attention score does not convey significant meaning in this context, and our primary focus is on the patterns in the attention maps, we felt that omitting the color bar would result in a less redundant figure.
> >
> > We hope these revisions help clarify the aspects of our work and provide a deeper understanding of our approach and its novelty. If these clarifications address your concerns, we would respectfully request that you reconsider the score.
> >
> > Thank you for your time and consideration.
> >
> > Sincerely,
> > The Authors

---

### Official Review · Reviewer_4bLg · 2024-11-04

**Soundness:** 2
**Presentation:** 2
**Contribution:** 2
**Rating:** 5
**Confidence:** 4

**Summary:**

The paper introduces a new task, Text-Guided Time Series Forecasting (TGTSF), which addresses limitations in traditional time series forecasting by integrating textual information as additional inputs. The authors propose TGForecaster, a model that uses cross-attention mechanisms to combine time series data with text, such as channel descriptions and news, enhancing forecasting accuracy. The paper presents four benchmark datasets, each designed to test different facets of this task, from periodic patterns to complex, event-driven fluctuations. Experimental results show that TGForecaster consistently outperforms baseline models, demonstrating the value of incorporating external text data in time series forecasting.

**Strengths:**

1. The paper is clearly written, with thorough explanations of model components and datasets.
2. TGForecaster is well-designed, with cross-attention mechanisms validated through comprehensive experiments across diverse datasets.
3. TGTSF demonstrates that textual context enhances forecasting accuracy, establishing a valuable foundation for future multimodal research in forecasting applications.

**Weaknesses:**

1. Limited novelty: The approach of using textual embeddings for time series forecasting, especially in financial domains, is not new. Prior work in stock forecasting, such as [1][2], has already explored combining news and other textual data with time series data to improve forecasting.
2. Compositional methodology: Since the channel descriptions are generated by large language models (LLMs), it would be logical and consistent to utilize embeddings directly from the LLMs used for text generation, rather than relying on a separate text embedding model.
3. Missing reference: The paper ignores the existing attempts in using textual embeddings for time series forecasting, such as [1][2]


[1] Sawhney, Ramit, Arnav Wadhwa and Shivam Agarwal. “FAST: Financial News and Tweet Based Time Aware Network for Stock Trading.” Conference of the European Chapter of the Association for Computational Linguistics (2021).
[2] Liu, Mengpu, Mengying Zhu, Xiuyuan Wang, Guofang Ma, Jianwei Yin and Xiaolin Zheng. “ECHO-GL: Earnings Calls-Driven Heterogeneous Graph Learning for Stock Movement Prediction.” AAAI Conference on Artificial Intelligence (2024).

**Questions:**

1. Could the authors clarify what unique insights or capabilities TGTSF adds to this field? Specifically, how does TGTSF extend beyond the use of composite news and channel descriptions?

2. Is there a specific reason why embeddings from the LLMs used for generation were not directly employed, instead of relying on an external text embedding model?

3.  Would the authors consider expanding the evaluation to include comparisons with other advanced multimodal models from financial or weather forecasting?

---

> ### Author Response · Authors · 2024-11-19
>
> - Novelty and Special insights:
>
>     We acknowledge that leveraging heterogeneous data to improve stock forecasting has been extensively studied, and it represents one of the key real-world applications targeted by our model. Unlike traditional methods, our approach eliminates the need for additional engineering or manual processing of textual data. Instead, the model automatically learns the correlation between specific events (e.g., news) and different time series channels (e.g., individual stocks), capturing their varying impacts.
>
>     Our approach is built on two major insights:
>
>     1.	**Semantic Enrichment with Text Embeddings**: By using text embeddings, our model incorporates rich semantic information, enabling it to uncover causal relationships between events and time series patterns. This is especially important since most events or news cannot be effectively represented as simple numerical time series.
>
>     2.	**Channel-Text Correlation via TGCI**: News is often not explicitly aligned with specific time series channels. To address this, we leverage the properties of text embeddings to compute correlations between channels and dynamic news using the inner product of channel descriptions and news embeddings—a mechanism we term Text Guided Channel Independence (TGCI). This allows the model to determine the relevance of news to specific channels without requiring explicit alignment or manual preprocessing.
>
>     With this approach, there is no need to transform textual news into numerical representations or manually analyze with domain knowledge and rephrase it for each channel. Instead, the text can be directly compressed into embeddings and fed into TGForecaster, enabling a fully end-to-end text-guided time series forecasting solution.
>
>     For further details, please refer to the Novelty section in the common response.
>
> - Why use dedicate text embedding:
>
>     Text embedding model is an encoder model that compress the input text as a vector in semantic space. Thus, we can calculate the similarity/correlation of embeddings with inner product. This is the key to every of our unique design of TGForecaster, (see paper Section method and common reply Section Novelty).
>
>     However, GPT is a decoder next token prediction only model, it is trained to generate next word token in autoregressive way. Thus, the embedding of it is not specifically trained to represent the text semantic. For now, it may not be a good idea to directly use it replace the text embedding. The use of text embedding is further detailed in the "Use of Text" section of the common response.
>
>     But of course, it may be a good idea if we directly integrate large language model as a part of our model, but this requires further design and proof of concept. We will explore this idea in the future work!
>
>     We have to emphasize that GPT is not a part of our model now. And it is simply used to generate various text as our dataset. Or in other words, if we have a dataset that already have text and its corresponding time series, we do not even bother to use the GPT, like the Steam dataset we used.
>
> - Analysis on two mentioned paper:
>
>     Both methods utilize text embeddings to help time series models understand the semantics of text, but each (including TGTSF) targets different tasks and serves distinct purposes.
>
> 	- ECHO-GL focuses on modeling dynamic graph relationships with the aid of text embeddings. While it shares a similar idea of computing correlations between keywords and stock descriptions, this method primarily uses these embeddings to construct a knowledge graph. However, it relies heavily on predefined graph-building algorithms, which may limit its generalization ability across diverse datasets or scenarios.
> 	- FAST employs BERT to encode textual information but appears to use these embeddings primarily as supplementary input, passing them through an LSTM model without leveraging the intrinsic properties of the embeddings. Additionally, this work is framed as a ranking task, distinct from TGTSF’s focus on causal forecasting.
>
>     Both are inspiring works, and we will include them in the related work section of the final version to better highlight our novelty.
>
> - Why Not Apply TGTSF to the Financial Domain?
>
>     Financial forecasting is indeed a potential use case for TGTSF. However, a key challenge is that news affecting the stock market is often not timely. By the time such news is published, its impact on the market may already have occurred, making it difficult for the model to learn robust cause-and-effect relationships between the text and the time series data.
>
>     Additionally, financial datasets are notoriously challenging to obtain due to intellectual property restrictions and access limitations. These constraints require further investigation into the availability and quality of finance-related data sources before fully exploring this application area.

---

> > ### Comment · Reviewer_4bLg · 2024-11-22
> >
> > While I appreciate the authors' efforts in addressing my concerns, the novelty of the method remains unclear. The proposed TGCI mechanism is presented as a key differentiator; however, the explanation does not convincingly establish how this approach significantly advances beyond a series of existing methods like FAST, TEXT, and GPT4MTS [1], which also combines textual and time-series data using text embeddings.
> >
> > Regarding the use of dedicated text embeddings, the authors’ justification is unconvincing. Decoder-only embeddings, such as those from GPT models, have already been adopted in prior work and shown to be effective in various tasks, including recommendation systems [2]. The authors have not addressed why these proven embeddings were not considered or compared to the dedicated embeddings used in TGForecaster.
> >
> > The reliance on LLM-generated text also raises additional concerns. The response does not clarify how variations in the quality of generated text might impact the results or whether alternative text generation methods were tested. Furthermore, if TGForecaster depends heavily on LLM-generated captions, its distinction from multi-modal LLMs like TimeLLM becomes blurred. The lack of clarity on this point weakens the argument for the method's uniqueness.
> >
> > The underperformance of TimeLLM on the weather dataset is another unresolved issue. While simpler models like DLinear outperform TimeLLM, the authors provide no insight into the underlying reasons for this discrepancy. This omission raises questions about the robustness of the assessment across diverse datasets.
> >
> > While I appreciate the authors' response and the additional details provided, the explanation does not fully address my concerns, particularly regarding the novelty and robustness of the proposed method. The distinction from prior work, the justification for embedding choices, and the method's scalability remain unclear. Unless the authors provide a more detailed and thorough explanation addressing these concerns, I will consider this work to be incremental and adjust my score accordingly.
> >
> > [1] Jia, Furong, Kevin Wang, Yixiang Zheng, Defu Cao and Yan Liu. “GPT4MTS: Prompt-based Large Language Model for Multimodal Time-series Forecasting.” AAAI Conference on Artificial Intelligence (2024).
> > [2] Hu, Jun, Wenwen Xia, Xiaolu Zhang, Chilin Fu, Weichang Wu, Zhaoxin Huan, Ang Li, Zuoli Tang and Jun Zhou. “Enhancing Sequential Recommendation via LLM-based Semantic Embedding Learning.” Companion Proceedings of the ACM on Web Conference 2024 (2024): n. pag.

---

> > > ### Author Response · Authors · 2024-11-23
> > > **Response #1**
> > >
> > > Thank you for your feedback and insightful follow-up questions. It seems there may be some misunderstanding of our work, so we would like to provide further clarification to address your concerns.
> > >
> > > - Large Language Model is NOT Part of Our Model
> > >
> > > We notice this is a common misunderstanding, and we want to clarify that **GPT is not part of our model, nor does our model rely on any capabilities of large language models.**
> > >
> > > GPT was used solely to generate diverse and correlated text for the weather-captioned dataset. This allowed us to simulate a time series dataset that inherently includes highly correlated textual weather reports. Within the context of our paper, the weather-captioned dataset is treated as an off-the-shelf TGTSF dataset with aligned text and time series data. The dataset generation process itself is a background effort, and the detailed information we provide about creating this dataset is purely for transparency and reproducibility, just as we do for our other datasets.
> > >
> > > The TGTSF task focuses on scenarios where the time series data naturally comes with correlated text, and the model’s goal is to extract causal relationships between the text and time series patterns. All the datasets we use—whether simulated or collected from the real world—are designed to meet this criterion. Beyond the weather-captioned dataset, we also include the Steam dataset, which directly uses raw text gathered from media sources. In this case, the text embeddings are passed directly to the model without involving GPT or any large language model. This demonstrates that our pipeline is inherently designed to operate without relying on LLM-generated captions.
> > >
> > > For further details on why we chose the weather dataset and the rephrasing process, please refer to the discussion with Reviewer GhyY.
> > >
> > > - Why Don’t We Use Large Language Models (LLMs) as Part of Our Model?
> > >
> > > Since LLMs are not involved in any part of our model’s training or inference process, the question of “why not use existing LLM embeddings instead of a separate text embedding model” becomes irrelevant. Instead, we focus on comparing using dedicate LLM and dedicate text embedding model:
> > >
> > > 1. **End-to-End Pipeline**: Our goal is to build an end-to-end system where text directly guides time series forecasting. The model learns to align time series patterns with the semantics of the input text casually. With existing correlated text, we do not need LLMs to rephrase or generate text and instead compress the text directly into embeddings to feed into our model. This ensures a streamlined pipeline tailored to our task.
> > > 2. **Let the models do what they are best at**: The text embedding model we use is specifically trained to map text into a vector space where semantically similar texts are represented by closely aligned vectors. While LLM embeddings can also represent text, they are optimized for tasks like next-token prediction rather than semantic compression. Using a task-specific text embedding model is more natural and effective for our purpose. For comparison, it’s like using a CLIP model for image retrieval instead of embeddings from an image generation model. While both contain information about images, the task-specific design of CLIP makes it more efficient and accurate.
> > > 3. **Avoiding Redundant Information:** LLM embeddings are designed to generate text and therefore include positional and token-specific information that is unnecessary for our task. This additional information can introduce noise, reducing the effectiveness of the semantic space representation required for causal alignment between text and time series.
> > > 4. **Robustness to Text Variations:** Text embedding models are trained to focus on semantics, filtering out irrelevant variations in expression. For instance, sentences like “The morning is clear and it will turn to rain in the afternoon” and “The day starts with clear skies and ends with evening rain” would produce similar embeddings in a dedicated text embedding model. In contrast, LLM embeddings are sensitive to textual expression, making them less robust for semantic alignment.
> > > 5. **Computational Efficiency:** LLMs are typically much larger and more computationally intensive than text embedding models. In scenarios where text already exists—as assumed in our task—using a text embedding model is far more efficient. This is especially important for time-sensitive applications like stock market analysis or control simulations, where rapid processing is crucial.
> > > 6. **Flexibility in Embedding Dimensions:** Advanced text embedding models, such as those trained with Matryoshka embedding techniques [1], allow flexibility in choosing embedding dimensions without retraining the model. For instance, OpenAI embeddings [2] can be truncated to 512 dimensions while preserving normalized representations. This enables a trade-off between model size and semantic detail, a property not typically available in LLMs since it’s unnecessary for their tasks.

---

> ### Author Response · Authors · 2024-11-23
> **Response #2**
>
> 7. **Limited Performance of LLMs for Time Series:** Recent studies [3] suggest that directly applying LLMs to time series forecasting offers limited performance improvements while significantly increasing computational overhead. This indicates that LLMs are not yet optimized for this type of task. Instead, our work provides an alternative approach by focusing on causal alignment between text and time series patterns.
>
> Given these reasons, we chose a dedicated text embedding model for our task. However, we recognize that integrating LLMs into our pipeline in an efficient and effective manner is an interesting direction for future work, and we aim to explore this further.
>
> [1] Introduction to Matryoshka Embedding Models https://huggingface.co/blog/matryoshka
> [2] OpenAI Embedding User Guide https://beta.openai.com/docs/guides/embeddings
> [3] Are Language Models Actually Useful for Time Series Forecasting? https://arxiv.org/pdf/2406.16964
>
> - Novelty of Our Work
>
> As elaborated in the common response, using text embeddings is not the novelty of our work but a natural and appropriate design choice. This is similar to how using the transformer was not the novelty of the BERT model; instead, its correct and insightful application made BERT groundbreaking. While there is existing work that employs text embeddings for various tasks with different insights, this does not diminish the significance of our contributions.
>
> As conclusion, our key novelty lies in two main aspects:
> 1. Defining a New Task: We introduce the text-guided time series forecasting (TGTSF) task, an inherently novel problem with broad applicability in real-world scenarios.
> 2. Designing TGForecaster: We propose TGForecaster as a model to validate this task. Instead of simply the text embedding as an extra information source and simply concatenating or adding it to the time series embedding [GPT4MTS, FAST], TGForecaster is built with specific insights into text embeddings, leveraging the properties of the semantic space to align text with time series patterns both robustly and efficiently. This design is unique and has not been explored in prior work. Our novelty does not stem from the use of text embeddings themselves but from how we apply them within the TGTSF framework.
>
> Further, it is not necessary to use the text embedding for our task and this is still and open question for the follow researchers of TGTSF.
>
> - TimeLLM’s Performance Issues
>
> We observed similar issues with TimeLLM during our experiments. Using the official code repository, we trained TimeLLM on our datasets without further modification. Not only did it struggle on the weather-captioned dataset, but it also performed significantly worse than DLinear on the toy dataset. This indicates that TimeLLM may lack the capability to handle the dynamic and causal time series patterns present in our datasets. In contrast, DLinear was able to capture basic patterns in the time series, maintaining some degree of generalization on our data.
>
> Additionally, the general capability of TimeLLM is questionable, as highlighted in [3]. However, investigating why it underperforms is beyond the scope of our work. Our focus remains on introducing and validating the TGTSF task and TGForecaster.

---

> > ### Author Response · Authors · 2024-11-25
> >
> > Dear reviewer,
> >
> > We hope the above response adequately address your aforementioned concerns. We are looking forward to further questions or concerns regarding our work during the remaining discussion period. If our clarifications address your concerns, we respectfully request you to consider revising the score.
> >
> > Sincerely,
> > Authors

---

> ### Author Response · Authors · 2024-11-27
>
> Dear Reviewer,
>
> We would like to sincerely thank you for your thoughtful feedback and suggestions on our manuscript. In response, we have updated the manuscript, incorporating several of your suggestions and addressing the concerns you raised. Specifically, we have added an updated ablation study that further clarifies the key mechanisms behind our design, and we have included a new background section that discusses text embedding techniques and related works. Additionally, we have added a section to highlight that our dataset is designed as an off-the-shelf text-time series dataset, requiring no further LLM captions, which we hope will help alleviate any potential confusion.
>
> We believe these revisions provide a clearer understanding of our approach and its novelty. If these clarifications address your concerns, we would respectfully request that you reconsider the score.
>
> Thank you for your time and consideration.
>
> Sincerely,
> The Authors

---

> > ### Comment · Reviewer_4bLg · 2024-11-27
> >
> > Thank you for the detailed clarification and for addressing the concerns regarding the use of LLMs and text embedding models. However, I remain unconvinced by the comparison of dedicated textual embeddings with LLMs, as the authors overlook the demonstrated strengths of LLM-generated contextual semantic embeddings. LLMs are widely adopted across various domains and have shown better performance in tasks requiring rich semantic representation. Even in this work, the generated text from LLMs could enhance the performance, proving that the knowledge embedded in LLMs could benefit the task here.
> >
> > The current approach appears limited by not leveraging the full potential of LLMs but a data generator, especially given their proven effectiveness in embedding tasks. Incorporating or benchmarking against LLM embeddings could provide stronger empirical evidence for the claims made and further validate the proposed method's distinctiveness and advantages.
> >
> > While I appreciate the additional insights and clarifications provided, the response does not address my core concerns, and I will maintain my score.

---

> > > ### Author Response · Authors · 2024-12-02
> > >
> > > Dear Reviewer,
> > >
> > > To address your concerns, we have further explained our design insights. Also, following your suggestion, we conduct a ablation study to benchmark the effectiveness of generative large language and the finetuned text embedding model. The result turned out to match our insight perfectly - the embedding directly from generative language model indeed brings performance boost comparing to the model that uses time series input alone. However, it still performs not as good as the embedding model that is further finetuned on text similarity task both in effectiveness and efficiency. Finally, we also provide a analogy to further help you understand why using text embedding model is a more natural and efficient design, particularly in our model.
> > >
> > > With the above information provided, we hope your concerns are adequately and evidently addressed. As the discussion period is coming to an end, we are looking for your feedback. If our clarifications address your concerns, we respectfully request you to consider revising the score.
> > >
> > > P.S. This is really a constructive discussion and we thank the reviewer for the effort.

---

> ### Author Response · Authors · 2024-11-27
>
> We appreciate your follow-up response and thoughtful feedback.
>
> In our work, we do not aim to compare dedicated textual embeddings with large language models (LLMs). **The focus of our research is not to determine which method is better**, but rather to highlight our design choice in selecting a text embedding model as a **feasible** and effective means of extracting semantic information. Our experimental results support this design choice, demonstrating that our approach can (but not only) work effectively with text embeddings.
>
> One of our untold motivation is to explore a method that does not follow generative LLM+X paradigm. We demonstrate a pipeline that does not rely on large language models and instead processes raw text directly, without requiring summarization, captioning, or reformatting via LLMs. Our results show that, with text embeddings, our model can robustly handle text input and achieve strong performance.
>
> Further in the dataset preparing, we do not leverage the advanced capabilities of LLMs, such as reasoning or generation; we simply use GPT to convert structured data into diverse natural language without further analysis, such as simply converting *{wind_dir: west, condition: sunny}* to sentences as *It will be clear tomorrow with breeze from the west.* All these can be done with programmed manner, but we want the expression diversity so that we choose to use GPT for faster generation. Therefore, it is not accurate to claim that our work demonstrates the advantages of LLMs for this task.
>
> Further, we would like to align our understanding of "large language model." By definition, the LLMs are machine learning models that designed to solve downstream natural language tasks which is not limited to language generation but also include text embedding. Many embedding models are also large and directly shares pretrained backbone of generative language model, such as *OpenAI’s text-embedding-3 model is actually a full GPT3 backbone that having been trained on same extensive datasets but with a head specially finetuned for downstream text similarity task, and it definitely and originally classified as Large Language Model by OpenAI*. So we want to know what does reviewer refer to by "LLMs". Does this refer to all pre-trained language models, including BERT and RoBERTa, or is it specifically meant to refer to generative language models (e.g., GPT, LLaMA), or it only has to be large enough? It seems that reviewer refer the LLMs as simply generative language models.
>
> Finally, as we mentioned in our previous responses, we acknowledge that integrating a large language model into the TGTSF framework is an interesting avenue for future work. To address your concern, we will provide additional experimental results using text embeddings generated by another generative language models (e.g., T5 or LLaMA2) in a follow-up response.
>
> We look forward to your response so that we can effectively address your concerns.

---

> ### Author Response · Authors · 2024-11-27
> **An interesting analogy to clear up confusion**
>
> To further clarify our perspective on text-embedding models and LLMs, we present an analogy that we believe will help the reviewer and AC understand our approach more clearly.
>
> In our work, the primary task involves extracting text similarity using cosine similarity, both in TGCI and in aligning text embeddings with time series embeddings. We can think of this task as needing to put a nail in the wall. The "dedicated text-embedding" model is like a hammer: it is specifically designed for the task of extracting text similarity, much like a hammer is designed for driving nails. On the other hand, generative large language models (LLMs) are like wrenches, they are trained to generate next token for dialogue. While their embeddings do indeed contain semantic information and one could technically use a wrench to hammer a nail into the wall, using a hammer for this task is natural, efficient, and safe—it is exactly what the hammer is designed for. On the contrary, if our goal were to enable dialogue capabilities in the model, we would choose to use an LLM, much like using a wrench to tighten a bolt. The decision is not about which tool is "better," but rather about selecting the most appropriate tool for the task at hand. It is a matter of using the right tool for the specific functionality we aim to achieve.
>
> **So, there should be no drawback in using text-embedding models when they are applied in the correct and effective way, i.e. text similarity. We applaud the innovative use of wrenches for creative tasks, but only when no better tool is available.**
>
> Similarly, our work stands apart from other time series analysis models that also use text embeddings. While some models use text embeddings as an added source of information (analogous to using the hammer as a paperweight), we use the hammer for its intended purpose—extracting text similarity—yielding more natural and effective results. Of course it can work as a paperweight and may works well, but it is not what it meant to be. Again, **we do encourage such innovation, but this should not be that reason that using the hammers properly is a bad and unwelcoming thing.**
>
> We think this simple analogy clearly illustrates our position and reinforces the value of using text-embedding models properly. We hope it helps to clear up any confusion and fosters further constructive discussion.

---

> ### Author Response · Authors · 2024-11-29
> **Simple benchmark between generative LLM embedding and finetuned text embedding #1**
>
> Dear reviewer,
>
> To further address your concern, we have conducted additional experiments to show a toe to toe comparison between the generative language (hopefully the LLMs you mentioned) and a "dedicate" text embedding model that is directly fintuned on the same generative model. **TLDR: the model that uses finetuned embeddings performs better than the ones that use the generative model embedding directly.** We have will detail our experiment setup and results below.
>
> We use the T5 [1] as the text-to-text generative model. Specifically, we directly use the `T5-base` [2] model from huggingface's transformers library to make sure the fairness. We extract the last_hidden_state from its encoder as the embedding of our input sentence as previous multimodal model such as Muse [7] and suggested in [5, 6].
>
> For comparison, we use the Sentence-T5 [3] text embedding model, which is directly fintuned on the original pretrained T5 model for text similarity tasks. We use the `sentence-t5-base` [4] model from huggingface's sentence-transformers library same as `mpnet` and `MiniLM`.
>
> It worth note that, these two model share the **exactly the same backbone architecture** and the only difference is the decoder are designed for and finetuned on different downstream tasks. The sentence-T5's encoder is initilized with T5's encoder parameters and further finetuned as described in [3]. So this provide a fair comparison between the raw embedding from generative model and the embedding that is specifically finetuned for text similarity tasks.
>
> We conduct experiment on weather-captioned medium dataset with two horizons, all model hyper-parameters are the same.
>
> |     | T5-base | Sentence-T5-base | T5-base @ 5epoch | Sentence-T5-base @ 5epoch | MiniLM | mpnet | PatchTST |
> | --- | - | - | - | - | -- | ----- | -------- |
> | 96  | 0.213   | 0.187 | 0.281    | 0.235    | 0.186  | 0.196 | 0.252    |
> | 192 | 0.264   | 0.216      | 0.321     | 0.280    | 0.214  | 0.216 | 0.304    |
>
> We not only report the final performance to demonstrate the *effectiveness* of these embedding, but also report the performance after 5 epochs to show the convergence speed for *efficiency*. The final result shows that the model uses generative embedding from `T5-base` performs not as good as the specially finetuned `Sentence-T5-base` text embedding model or any other text embedding models. However, it is still better than the performance of PatchTST which access no external textual information at all. This result perfectly aligns with our claim in the previous response that the **embedding from generative model certainly will bring benefits comparing to no textual information at all, but it is not as good as the specially finetuned text embedding model**, i.e. wrench is better than nothing but not as good as a hammer. Interestingly, the `sentence-t5-base` model sometimes even performs better than the `mpnet`, which shows that it is not a random choice and T5 backbone is indeed capable of handling text similarity tasks after finetuning.
>
> At the checkpoint of 5 epoches, we find that model that uses `sentence-T5-base` text embedding performs better than the one uses generative embedding from `T5-base` which indicate a batter convergence speed. **This aligns with our claim that text-embedding model creates a better semantic space specifically for text similarity tasks. Thus, make our TGForecaster do not have to learn the semantic space from scratch. While, the `T5-base` embedding contains some semantic information, it still need further learning to adapt to the text similarity tasks.**
>
> Finally, the `T5-base` model has about 223M parameters while the `Sentence-T5-base` model has about 110M parameters because it does not need the large decoder for text generation. This makes the "text embedding" model as `sentence-T5-base` **not only delivers better performance but also more efficient in terms of model size and inference speed**, as we argued in previous response.
>
> As we explained before, we never overlook generative LLM as a powerful model that can benefit many downstream tasks. However, *in our task and model design*, we only need the embedding to represent text semantic for similarity calculation instead of text understanding or generation. Thus, the text embedding model is an efficient and effective choice *for our model*.
>
> Here are some other cases that someone asks about extracting embedding directly from generative model and the answer is always the same: why not use text embedding model. [5, 6]
>
> Directly extracting very good semantic embedding directly from generative models is still a *valuable and ongoing* research topic. Although our "benchmarking against LLM embeddings" is not thorough (since it is not in the scope of our work), it still provide strong empirical evidence to support our design.

---

> ### Author Response · Authors · 2024-11-29
> **#2**
>
> We hope our **analysis, analogy and experiment results** altogether can help reviewer understand why we choose embedding model in our TGForecaster and address the concern.
>
> [1] Exploring the Limits of Transfer Learning with a Unified Text-to-Text Transformer, https://arxiv.org/abs/1910.10683
>
> [2] https://huggingface.co/google-t5/t5-base
>
> [3] Sentence-T5: Scalable Sentence Encoders from Pre-trained Text-to-Text Models, https://arxiv.org/abs/2108.08877
>
> [4] https://huggingface.co/sentence-transformers/sentence-t5-base
>
> [5] https://discuss.huggingface.co/t/how-to-use-t5-for-sentence-embedding/1097/7
>
> [6] https://stackoverflow.com/questions/76926025/sentence-embeddings-from-llama-2-huggingface-opensource
>
> [7] Muse: Text-To-Image Generation via Masked Generative Transformers, https://arxiv.org/abs/2301.00704
>
> P.S We still think this is really an insightful and constructive discussion though it is not in the scope of our work. And we thank the reviewer for asking us to explore this direction.

---

### Note · Authors · 2025-01-23

I have read and agree with the venue's withdrawal policy on behalf of myself and my co-authors.